# Evolutionary dynamics of circular RNAs in primates

**Gabriela Santos-Rodriguez[1,2], Irina Voineagu[3], Robert J Weatheritt[1,2]\***

[1]EMBL Australia, Garvan Institute of Medical Research, Darlinghurst, Australia; [2]St. Vincent Clinical School, University of New South Wales, Darlinghurst, Australia; [3]School of Biotechnology and Biomolecular Sciences, University of New South Wales, Sydney, Australia

**Abstract** Many primate genes produce circular RNAs (circRNAs). However, the extent of circRNA conservation between closely related species remains unclear. By comparing tissue-specific transcriptomes across over 70 million years of primate evolution, we identify that within 3 million years circRNA expression profiles diverged such that they are more related to species identity than organ type. However, our analysis also revealed a subset of circRNAs with conserved neural expression across tens of millions of years of evolution. By comparing to species-specific circRNAs, we identified that the downstream intron of the conserved circRNAs display a dramatic lengthening during evolution due to the insertion of novel retrotransposons. Our work provides comparative analyses of the mechanisms promoting circRNAs to generate increased transcriptomic complexity in primates.

**\*For correspondence:**
r.weatheritt@garvan.org.au

**Competing interest:** The authors declare that no competing interests exist.

## Introduction

An important question in biology is how has the complexity of biological systems expanded while the number of protein-coding genes has remained mostly stable. Through decades of research, it has been shown that increased biological complexity has arisen in part by the dynamic generation of unique cell-specific transcriptomes, and as a consequence of the highly versatile programs of gene expression (*Brawand et al., 2011*; *Cardoso-Moreira et al., 2019*). However, studies of tissues across distant animal lineages have shown that gene expression is highly conserved between the same tissues in different species (*Barbosa-Morais et al., 2012*; *Brawand et al., 2011*; *Cardoso-Moreira et al., 2019*; *Merkin et al., 2012*; *Reyes et al., 2013*). Hence, gene expression alone is unlikely to explain the heterogeneous expansion in complexity (as defined by the number of cell types) across vertebrate evolution. Instead, it is becoming increasingly evident that the plethora of post-transcriptional mechanisms (*Cheetham et al., 2020*; *Fiszbein et al., 2019*; *Gueroussov et al., 2017*; *Ha et al., 2018*; *Ha et al., 2021*; *Mattick, 2018*) capable of greatly expanding transcriptomic diversity also underlies these advances.

Among these, an intriguing class produced by pre-mRNA processing are circular RNAs (circRNAs) (*Zhang et al., 2013*; *Memczak et al., 2013*; *Li et al., 2018b*; *Gokool et al., 2020a*). These RNAs can regulate protein localization (*Liu et al., 2019*), miRNA functionality (*Piwecka et al., 2017*), and a range of other processes (*Li et al., 2018a*; *Gokool et al., 2020a*), enabling increased regulatory complexity, especially in the immune and nervous systems (*Gokool et al., 2020b*; *Li et al., 2017*; *Liu et al., 2019*; *Piwecka et al., 2017*). CircRNAs form by back-splicing whereby an exon's 3′-splice site is ligated to an upstream 5′-splice site forming a closed circRNA transcript (*Barrett et al., 2015*; *Starke et al., 2015*). Back-splicing occurs both co- and post-transcriptionally and is facilitated by inverted repeat elements that promote complementarity between adjacent introns favoring circRNA formation over linear splicing (*Ivanov et al., 2015*; *Jeck et al., 2013*; *Liang and Wilusz, 2014*; *Zhang et al., 2014*). These RNA-RNA interactions can be facilitated by RNA-binding proteins, such as Quaking (*Conn et al., 2015*), that help stabilize the hair-pin structure promoting circRNA formation.

The production of circRNAs can also arise due to the perturbed expression of trans-factors and the inhibition of the core splicing machinery (*Aktaş et al., 2017*; *Liang et al., 2017*). These spuriously produced circRNAs are maintained as their circular shape protects them from the activity of cellular exonucleases (*Gokool et al., 2020b*). In contrast, the variable usage of cis-regulatory elements in exons and flanking introns can be selected to promote circRNA expression in a cell-type, condition- or species-specific manner (*Irimia and Blencowe, 2012*; *Nilsen and Graveley, 2010*). Changes in circRNA expression may therefore represent a major source of species- and lineage-specific differences or error-prone mis-splicing. To provide insight into this quandary, here we describe a genome-wide analysis of circRNAs across physiologically equivalent organs from primate species spanning 70 million years of evolution. Our analysis uncovers extensive evidence of species-specific circRNAs that display no evidence of conservation even across relatively short evolutionary time periods. However, we also identify a small subset of circRNAs that are conserved across tens of millions of years displaying increased inclusion rates across evolutionary time. Our analysis comparing conserved circRNAs to species-specific circRNAs reveals that these circRNAs are flanked by newly inserted transposons that correlate with circRNA genesis and extend intron downstream of circRNA. Overall, our results identify evidence of circRNA conservation within closely related species and identify a reoccurring mechanism that correlates with circRNA genesis facilitating the expansion of transcriptomic complexity of primate cells.

## Results

### A core subset of circRNAs show conserved expression signatures but most are species-specific

To address the outstanding questions about the conservation and functional importance of circRNAs, we collected transcriptomic (RNA-seq) data (*Peng et al., 2015*; *Pipes et al., 2013*) from across nine tissues from eight primate species, consisting of three old-world monkeys, two hominoids, two new-world monkeys, and one prosimian (*Supplementary file 1*). These species were chosen on the basis of the quality of their genomes and their close evolutionary relationships enabling the evaluation of transcriptome changes between species ranging from <3 million years to >70 million years (see *Figure 1A*). For each species, we considered all primate-conserved internal exons as potential origins of back-spliced junctions (BSJs) with no restrictions on backward exon combination. Only canonical and annotated splice sites were used in analysis. RNA-seq reads were mapped to exon-exon junctions (EEJs) to determine 'percent spliced in' (PSI) for all circRNA with respect to the linear transcript. We also calculated PSI values for linear splicing of each internal exon and transcript per million (TPM) values to estimate gene expression. Orthology relationships between genes and exons were established to enable direct cross-species comparisons.

The circRNA analysis was done using Whippet because, according to our benchmarking results (see Materials and methods for details), it is an accurate and fast circRNA quantification tool. Our analysis of both simulated and collected RNA-seq data found that Whippet has a low false positive rate (<2%, see Materials and methods for details), which is in line with other methods (*Szabo et al., 2015*, *Gokool et al., 2020a*), a high rate of circRNA identification even at low read depths (~90% ; *Figure 1—figure supplement 3C*) and is faster (~69 min) with less computational overhead (<3 GB of memory on a single core) than other highly cited circRNA algorithms we compared with (CIRC-explorer3 [*Ma et al., 2019*], CIRIquant [*Zhang et al., 2020*], and find_circ [*Memczak et al., 2013*]; *Figure 1—figure supplement 3A and B*).

We initially explored the expression relationships within our datasets using hierarchical clustering and Pearson's correlations to determine the gene expression relationships between orthologous genes (see Materials and methods). In agreement with previous results (*Brawand et al., 2011*; *Merkin et al., 2012*; *Barbosa-Morais et al., 2012*; *Reyes et al., 2013*) from analysis across vertebrate species, a clear pattern emerged of tissue-specific conservation of gene expression (*Figure 1B*). This pattern suggests that most tissues possess a tissue-specific gene expression signature such that, for example, a liver-specific gene in chimp will likely also be liver-specific in lemur. In contrast to previous observations in vertebrates (*Merkin et al., 2012*), there are no clear species-specific exceptions to these patterns likely reflecting the closer evolutionary relationships studied.

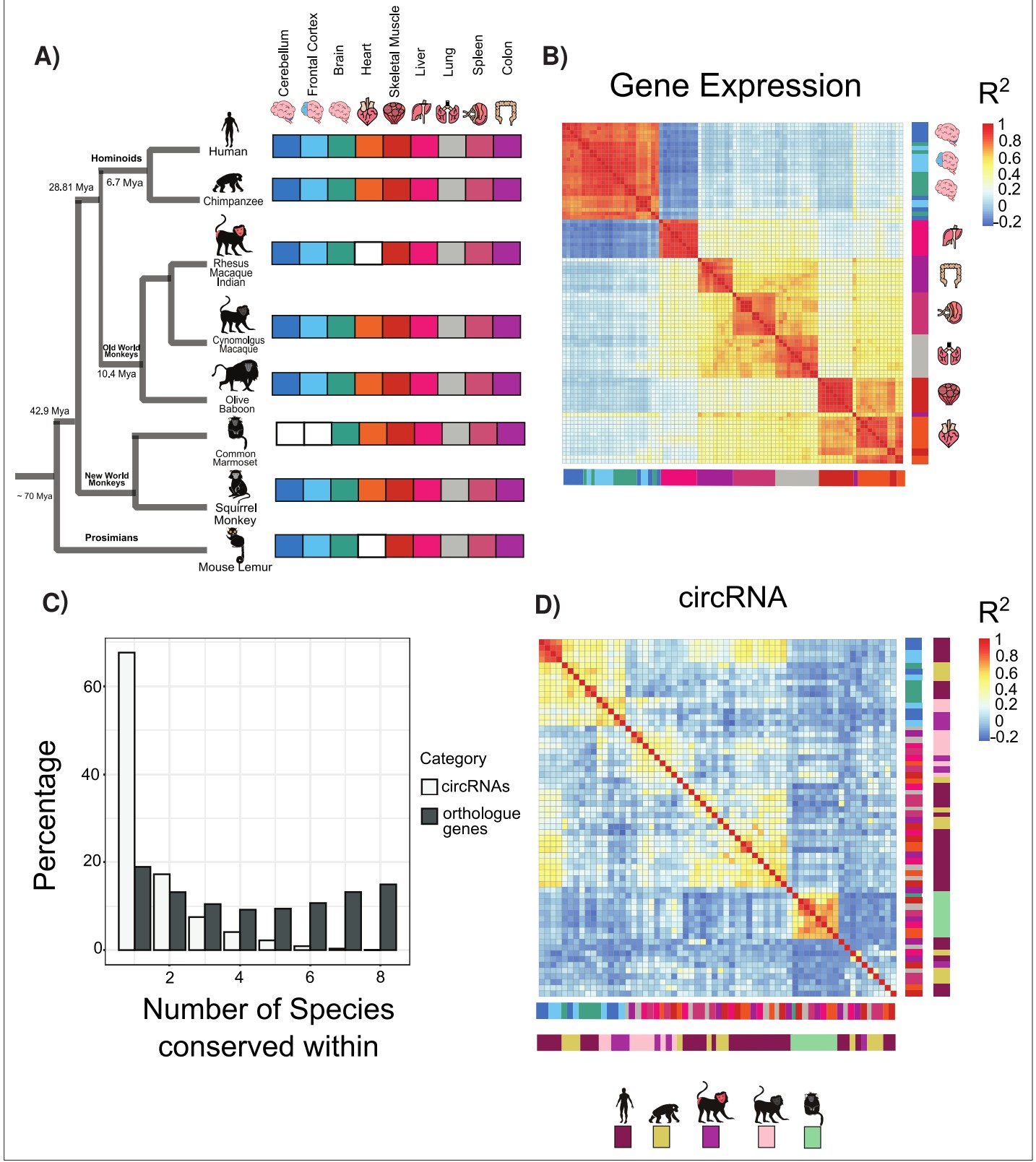

**Figure 1.** Circular RNA (circRNA) expression signatures are conserved in some tissues. (**A**) Phylogenetic tree of analyzed species with distance from human in millions of years (MYA) (divergence time according to TimeTree http://www.timetree.org/). Tissue datasets used in analysis identified on right with white squares denoting lack of dataset. (**B**) Clustering of samples based on expression values (transcripts per million). The variance of expression values was calculated, and the top 1000 most variable genes were used to calculate Pearson's correlation (n = 1000 genes in 88 samples). Red colors

*Figure 1 continued on next page*

*Figure 1 continued*

indicate high correlation between samples, and blue describes low correlation. Vertical and horizontal adjacent heatmaps describe tissues (see **A** for key). (**C**) Barplot showing conservation of circRNAs based on back-spliced junction and based on occurrence within orthologous genes. (**D**) Clustering of conserved circRNAs based on percent spliced in (PSI) values. Clustered using Pearson's correlation as in (**B**) (n = 149). Vertical and horizontal adjacent heatmaps describe tissues (inner heatmap; see **A** for key) and species (outer heatmap).

The online version of this article includes the following figure supplement(s) for figure 1:

**Figure supplement 1.** circRNAs expression and exons splicing patterns across primates species and in neuronal tissues.

**Figure supplement 2.** Validation of identified circRNAs using RNase R data.

**Figure supplement 3.** Benchmarking of Whippet for circRNA quantification.

To understand circRNA relationships between species, we performed an analogous pairwise clustering analysis using circRNA inclusion values. Replicates from the same tissue invariably clustered together. However, in contrast to gene expression, circRNA expression is segregated by species (*Figure 1—figure supplement 1A*). This suggests that despite all the exons studied being conserved across primates the majority of circRNAs showed species-specific expression with no orthologous circRNAs in other species (*Figure 1C*, ~67% are species-specific, n = 11,201). To evaluate the expression patterns of circRNA orthologs, we identified circRNAs with matched BSJs (see Materials and methods) conserved across ~45 million years of evolution. In this analysis, more complex patterns of circRNA conservation emerged with tissue-dominated clustering observed across all types of brain samples (*Figure 1D*) in line with previous observations (*Rybak-Wolf et al., 2015*; *Venø et al., 2015*; *You et al., 2015*). In contrast, for all other tissues circRNAs showed primarily species-specific clustering.

We next assessed if these changes may be explained by gene expression changes in the host gene. A comparison of genes containing conserved and species-specific circRNAs did not show any significant differences (*Figure 2—figure supplement 4A and B*, p=0.584 Wilcoxon rank-sum test), suggesting that differences between these subgroups are not driven by gene expression differences. We next evaluated if tissue-specific changes observed in the conserved circRNAs were due to tissue-specific gene expression or alternative splicing. Interestingly, genes containing conserved circRNAs neither displayed neural-specific gene expression (*Figure 1—figure supplement 1B*) or neural-specific alternative splicing changes (*Figure 1—figure supplement 1C*). This suggests that circRNA conservation and expression is independent of these regulatory layers.

We next investigated the genes containing circRNAs. Many orthologous genes consistently express circRNAs even if the precise BSJ is not conserved (*Figure 1C*). This phenomenon persisted across species with a median of 10 circRNAs detected per gene across tissues (*Figure 1—figure supplement 1D*). However, this circRNA production only occurred in a limited number of expressed genes (20.4% of orthologous expressed genes). This suggests that certain genomic areas are circRNA factories that are prone to produce large numbers of lowly expressed circRNAs.

These observations suggest that a core set of circRNAs show conserved tissue-specific patterns across neural tissues. However, the great prevalence of circRNAs showing species-specific expression indicates that the cis-regulatory or trans-regulatory environments may differ between even very closely related species to promote the species-specific production of circRNAs.

## Features of conserved circRNAs

Our analysis (*Figure 2A*) reveals clear subsets of several hundred circRNAs exhibiting highly conserved circRNA expression. The circRNA ERC1 and many other examples from our data (*Figure 2B*, *Supplementary file 2*, and *Figure 2—figure supplement 1A*) demonstrate that circRNA expression can be conserved for tens of millions of years.

To assess the phylogenetic distribution of circRNA across primates, we grouped them by PSI values requiring PSI ≥ 5 and at least five read support. Out of the approximately 56,000 internal exons with clear orthologs across primates, we identified a large set of circRNA expressing a 'species-specific' expression, as well as a set of ~773 'conserved circRNAs' that shared expression across at least human, chimp, and baboon (*Figure 2—figure supplement 1B and C*). Using our transcriptomic data, we found that a circRNA identified in human was approximately five times more likely to be identified in baboon than in lemur, in line with the closer phylogenetic relationship of human to baboon than human to lemur.

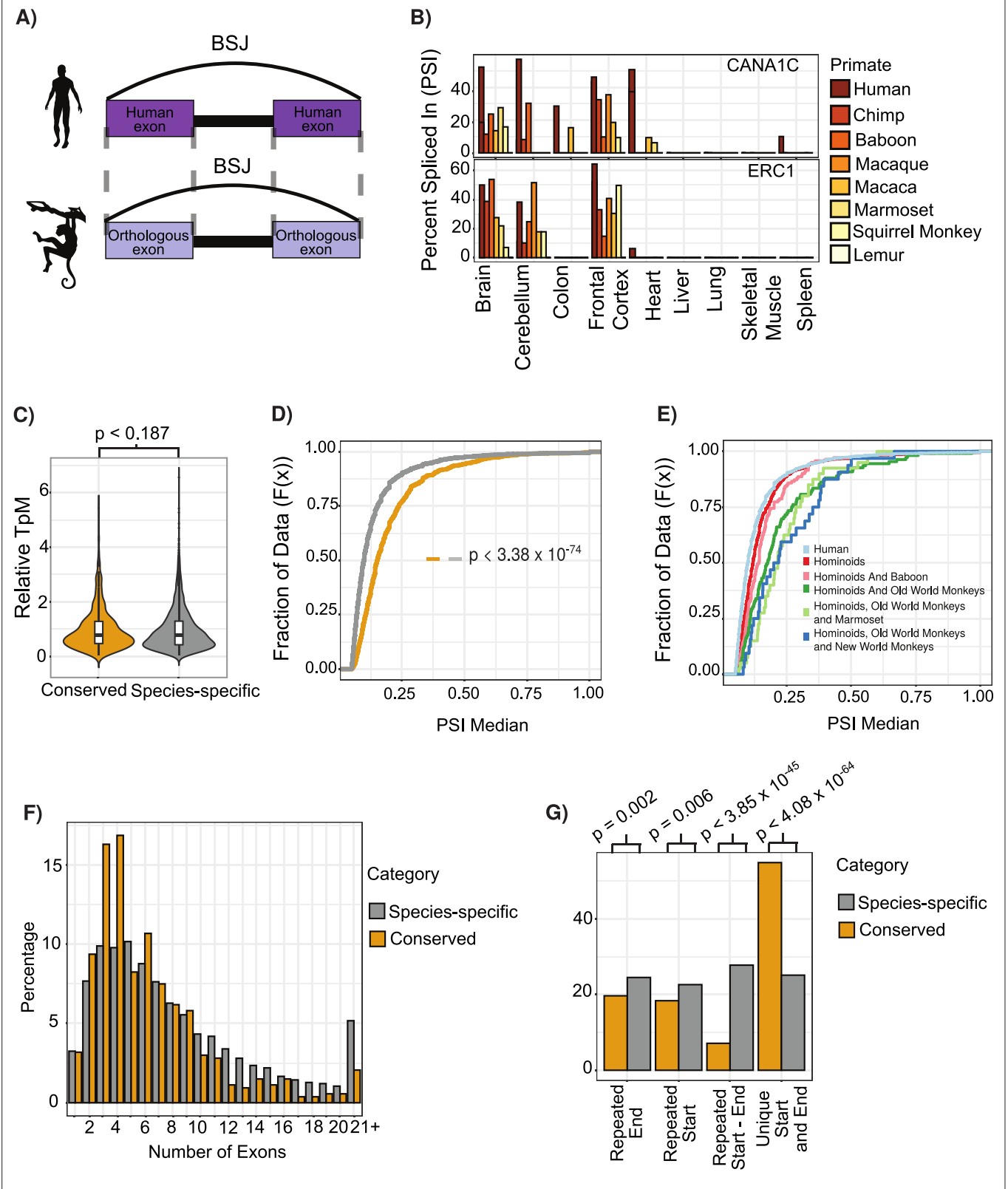

**Figure 2.** Features of conserved circular RNAs (circRNAs). (**A**) Schematic overview of identification of back-spliced junctions (BSJ) between species. (**B**) Percent spliced in (PSI) values for conserved circRNAs (top) CACNA1C_chr12:2504436–2512984 and (bottom) ERC1_chr12:1180540–1204512 across tissues and species analyzed. PSI values only calculated for circRNAs with more than five reads support. Gene name is indicated in top right-hand corner. (**C**) Violin plot describing relative expression levels of conserved and species-specific circRNAs. Violin plots show probability densities of the data

*Figure 2 continued on next page*

*Figure 2 continued*

with internal boxplot. Boxplot displays the interquartile range as a solid box, 1.5 times the interquartile range as vertical thin lines and the median as a horizontal line. p-Value calculated using Wilcoxon rank-sum test (p<0.187). TpM: transcripts per million. (**D**) Cumulative distribution plot of change in PSI values across all conserved (yellow) and species-specific (gray) circRNAs. A cumulative distribution plot describes the proportion of data (y-axis) less than or equal to a specified value (x-axis). Cumulative distribution $F(x)$, cumulative distribution function. p-Value calculated using Wilcoxon rank-sum test (p<3.38 $\times$ 10$^{-74}$). (**E**) Cumulative distribution plots of circRNAs with different levels of conservation, as defined by consistent observation of BSJ across species indicated. See (**D**) for description of cumulative distribution plot. (**F**) Barplot describing number of exons per circRNA for conserved and species-specific circRNAs. Exons are defined by Ensembl and must show evidence of expression (PSI >5 and > 5 reads support) in tissue analyzed. (**G**) Barplot describing uniqueness of start (5'-splice site) and end (3'-splice site) for conserved and species-specific circRNAs. p-Values calculated from Fisher's exact test (p<4.08 $\times$ 10$^{-64;}$; unique start and end – also see *Figure 2—figure supplement 3*).

The online version of this article includes the following figure supplement(s) for figure 2:

**Figure supplement 1.** Tissue-specific expression and conservation of circRNAs across primates species.

**Figure supplement 2.** Comparison of the expression of conserved and species-specific circRNAs across tissues samples.

**Figure supplement 3.** Overview of approach to identifying unique circular RNAs (circRNAs) for *Figure 2G* (see Materials and methods for details).

**Figure supplement 4.** Comparison of gene expression distribution of genes containing conserved and species-specific circRNAs across tissues samples.

To validate the quality of our identified circRNAs, we initially overlapped our data with circRNAs previously reported in circAtlas (*Wu et al., 2020*). This analysis found that 99.5% of the conserved circRNAs and 97.03% of species-specific circRNAs have been previously reported. Additionally, we verified our circRNAs dataset using RNase R data (see Materials and methods for details). This analysis of human data validated 82.7% of the conserved circRNAs (648 conserved circRNAs), despite these datasets not being from matched tissue samples (*Figure 1—figure supplement 2A*; see Materials and methods for details). To validate the conservation of our neuronal circRNAs, we next analyzed RNase R samples from different brain macaque regions. This analysis identified ~89% of the conserved circRNAs (324 conserved circRNAs;) (*Figure 1—figure supplement 2F*; see Materials and methods for details).

Initial analysis of conserved circRNAs revealed enrichment within neural tissues with over 70% showing consistent tissue expression across 30 million years of evolution (*Supplementary file 2*), in line with previous observations (*Rybak-Wolf et al., 2015*; *Venø et al., 2015*; *You et al., 2015*). Analysis of expression levels revealed no clear trends for increased expression of conserved circRNAs (*Figure 2—figure supplement 2A*, p<0.187, Wilcoxon rank-sum test vs. species-specific); however, these circRNAs did display increased inclusion rates or increased circRNA expression as compared to linear isoform (*Figure 2—figure supplement 2B*, p=3.38 $\times$ 10$^{-74}$, Wilcoxon rank-sum test vs. species-specific). Furthermore, this inclusion (or circularization) increased with the conservation age of the circRNA (*Figure 2E*, p=8.07 $\times$ 10$^{-19}$, Wilcoxon rank-sum test of hominoids vs. species-specific [human-specific]; p=2.14 $\times$ 10$^{-06}$, Wilcoxon rank-sum test of hominoids vs. shared until new-world monkeys). This suggests that over time these circRNAs are increasingly influencing the transcriptomic abundance of the linear isoform and the protein abundance of the gene.

Analysis of the exonic structure of conserved circRNAs showed that conserved circRNAs contain fewer exons (*Figure 2F*, *Figure 2—figure supplement 4C*, p = 2.23 $\times$ 10$^{-20}$, Wilcoxon rank-sum test) with a significant enrichment to contain 2–3 exons (p-value = 4.17 $\times$ 10$^{-08}$, Fisher's exact test), which is in line with observations from previous studies (*Ragan et al., 2019*). Conserved circRNAs also rarely overlap with other circRNAs (*Figure 2G*, p=4.08 $\times$ 10$^{-64}$, Fisher's exact test; see Materials and methods) displaying back-splicing at unique 5'- and 3'-splice sites. This indicates a tight control of the number of exons within a circRNA and the BSJs used.

## Conserved circRNAs have extensive downstream introns and are flanked by inverted repeat elements

To investigate the role of cis-regulatory elements within conserved circRNAs, we analyzed almost 150 features associated with circRNA formation including a multitude of trans- and cis-regulatory factors and all major groups of transposons (see Materials and methods and *Supplementary file 3*). To evaluate the influence of these features on defining conserved circRNAs, we used two background datasets (see *Supplementary file 2* and Materials and methods). The first is a background set of randomly

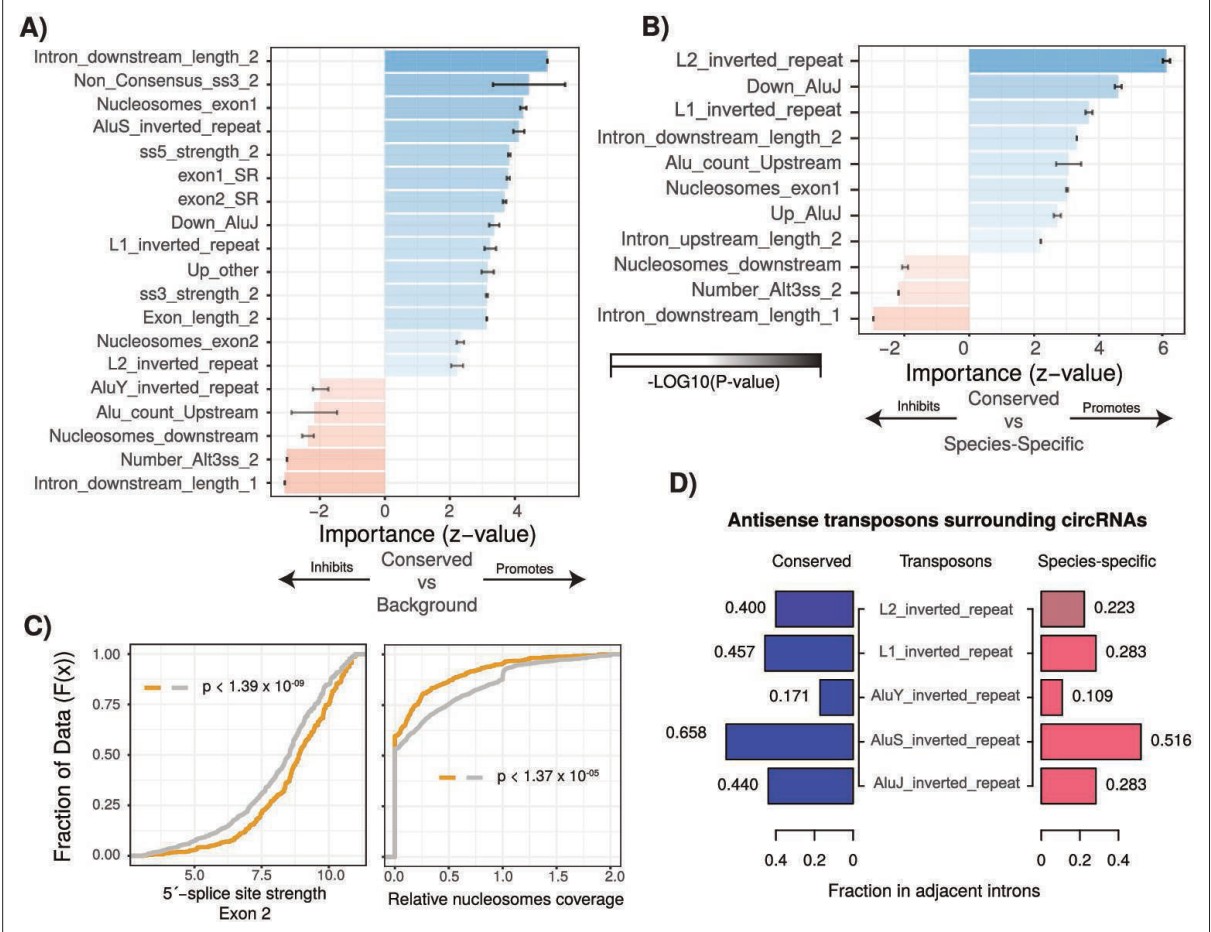

**Figure 3.** Characterization of cis- and trans-regulatory features of conserved circular RNAs (circRNAs). (**A**) Barplot describing feature importance for logistic regression model of conserved circRNAs compared to background. Colors represent positive or negative influence. Transparency reflects log10(p-value of z-statistic). Error bars represent standard error. '_1' is relative to first exon of circRNA and '_2' is relative to final exon of circRNA. ss3: 3′-splice site; ss5: 5′-splice site; Alt3ss: alternative 3′-splice sites. Inverted repeats are repetitive elements on opposite strands in introns adjacent to circRNAs. See ***Supplementary file 3*** for details of features. (**B**) Barplot describing feature importance for logistic regression model of conserved circRNAs compared to species-specific circRNAs. See (**A**) for plot interpretation and descriptions. (**C**) Cumulative distribution plots describing (left; p<1.39 × 10[-09]) 5′-splice site strength at final exon of circRNAs and (right; p<1.37 × 10[-05]) distribution of nucleosomes on intron downstream of circRNA. p-Values calculated by Wilcoxon rank-sum test and corrected for multitesting (Bonferroni). See ***Figure 2D*** for interpretation of cumulative distribution plot. (**D**) Pyramid plot showing the mean fraction of circRNAs with selected inverted repeat retrotransposon elements in adjacent introns.

The online version of this article includes the following figure supplement(s) for figure 3:

**Figure supplement 1.** Performance assessment of logistic regression model.

combined alternative (10 < PSI < 90) exons extracted from genes containing conserved circRNAs (background set). The second is the group of 'species-specific circRNAs' defined previously.

Using logistic regression combined with a genetic algorithm for model selection taking into account multicollinearity (see Materials and methods), we initially sought to determine the relative contribution of this diverse range of features in defining conserved circRNAs. After initially training our model on a subset of conserved and background circRNAs (80%), we next assessed its performance on the rest of 20% cirRNAs and observed a high average true-positive rate of 86.7% (AUC, area under the receiver operating characteristic [ROC] curve; ***Figure 3—figure supplement 1A***) for a model including 24 variables selected by feature analysis. This identifies a core set of 24 cis- and trans-regulatory features enriched within the conserved formation of circRNAs compared to our background set of introns (***Figure 3A and B***). This includes multiple features previously associated with conserved circRNAs, such as inverted repeat Alu elements (***Jeck et al., 2013***; ***Zhang et al., 2014***), as well as exon and intron length (***Ashwal-Fluss et al., 2014***; ***Ivanov et al., 2015***; ***Jeck et al., 2013***; ***Liang et al., 2017***).

We next used the same approach to determine features differentiating conserved and species-specific circRNAs. As expected, our model distinguished these categories less efficiently but was still able to achieve a true-positive rate of 65.4% (*Figure 3—figure supplement 1B*) driven by 12 features. Notable among these features was the depletion of nucleosomes in the downstream intron of the circRNA (*Figure 3—figure supplement 1D*, $1.57 \times 10^{-03}$, Bonferroni-corrected Wilcoxon rank-sum test [BH-Wilcox] vs. species-specific) and the presence of a more defined 3′-splice site at the final exon (p=$2.04 \times 10^{-03}$, BH-Wilcox vs. species-specific). Introns adjacent to conserved circRNAs also exhibited a significant enrichment for repeat elements (*Figure 3D*, all p<$1 \times 10^{-5}$, BH-Wilcox vs. species-specific) in particular inverted-repeat L1 and AluJ retrotransposons (:*Figure 3D*, L1: p<$1.22 \times 10^{-23}$| AluJ: p<$1.48 \times 10^{-18}$, BH-Wilcox). A further key distinguishing feature of interest was intron length. Conserved circRNAs exhibited shorter introns downstream of the first exon and an extended intron downstream of the final exon (*Figure 4A and B*). In species-specific circRNA, this adjacent downstream intron has a median length of 4624 nucleotides whilst in conserved circRNA the median is almost twice as long at 9923 nucleotides (*Figure 4B*, p<$1.07 \times 10^{-35}$, BH-Wilcox). Finally, when comparing the major drivers of both models, we noticed over 90% (11/12) of features overlapped between the models. This suggests that conserved circRNAs are an extreme continuum of species-specific circRNAs. Therefore, understanding the processes contributing to circRNA conservation may also provide insight into the genesis of circRNAs across species.

## Insertion of young transposons increases downstream intron length in conserved circRNAs

To investigate the evolutionary origins of the switch of conserved circRNAs from absence in prosimians and new-world monkeys to conservation within hominoids and old-world monkeys, we investigated the changes in intronic length for the orthologous introns between human (hominoids) and lemur (prosimians). In contrast to orthologous lemur introns, the human introns downstream of all identified circRNAs shows an almost fourfold expansion compared to background dataset of introns within circRNA containing genes (*Figure 4C*, p<$3.84 \times 10^{-23}$, Wilcoxon rank-sum) and the upstream adjacent intron (*Figure 4—figure supplement 1A*, p<$1.02 \times 10^{-10}$, Wilcoxon rank-sum). This difference is even greater in conserved circRNA, which display an almost twofold greater lengthening than species-specific circRNAs (or eightfold over background; *Figure 4C*, p<$3.84 \times 10^{-06}$, Wilcoxon rank-sum). These observations suggest that the expansion of the intron downstream of the circRNA may increase the proportion of back-splicing events increasing the likelihood of circRNA conservation.

To investigate the drivers of this intronic expansion, we aligned the lemur and human introns to identify regions novel to humans. This analysis revealed the insertion of novel transposons at almost double the frequency in introns associated with conserved circRNAs (*Figure 4D*, p<$5.48 \times 10^{-06}$, Wilcoxon rank-sum). Further evaluation of the retrotransposons revealed that this increase in length is driven by the novel insertion of AluJ and L1 elements (*Figure 4E*, AluJ: p<0.018; L1: p<$1.73 \times 10^{-04}$, Wilcoxon rank-sum). This retrotransposition is potentially facilitated by the depletion of nucleosome occupancy in these introns compared to other human introns (*Figure 3B*, p<$1.15 \times 10^{-07}$, BH-Wilcox). Together, this argues for the role of young transposons in creating longer intronic regions, which increases the time for RNA polymerase II to reach next canonical splice site and therefore increases likelihood of back-junction splicing to occur.

## Discussion

The evolution of circRNAs has been previously studied across extensive evolutionary time revealing poor conservation for the majority of circRNAs (*Rybak-Wolf et al., 2015*; *Venø et al., 2015*). Our approach is unique as it focuses on the conservation of circRNAs in very closely related species, enabling us to account for the rapid evolution of these RNAs. This increased resolution allowed us to compare conserved versus non-conserved circRNAs, enabling us to reveal two disparate facts about circRNA expression. Firstly, we observe extensive variation in the production of the vast majority of circRNAs between species. With circRNAs often expressed within the same orthologous genes even if BSJ is not conserved. Conversely, we identify a core set of over 700 circRNAs that are conserved across millions of years of evolution. These circRNAs have higher inclusion rates and show increased inclusion across evolutionary age. Both groups are related in the cis- and trans-regulatory features that correlate with circRNA formation such

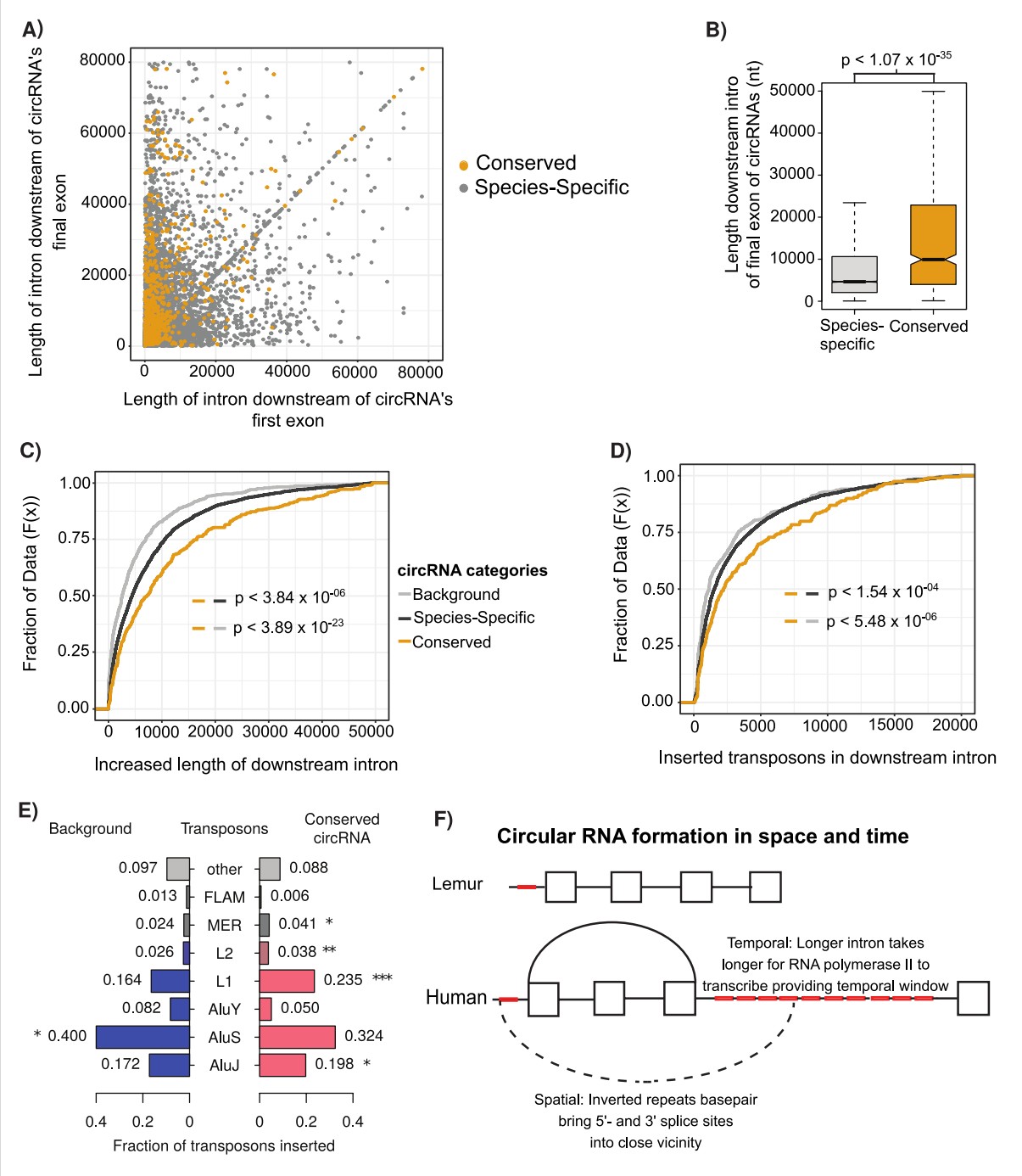

**Figure 4.** Conserved circular RNA (circRNA) downstream intron expanded during primate evolution. (**A**) Scatterplot of downstream intron length for conserved and species-specific circRNAs. (**B**) Boxplot describing lengths of intron immediately downstream of circRNA for conserved and species-specific circRNAs (see *Figure 2C* for description of boxplots). p-Values calculated by Wilcoxon rank-sum test and corrected for multitesting (Bonferroni). nt: nucleotide (**C**) Cumulative distribution plot of change of length of orthologous downstream introns of conserved, species-specific and background circRNAs from lemur to human (see *Figure 2D* for description of cumulative distribution plots). p-Values calculated by Wilcoxon rank-sum test and corrected for multitesting (Bonferroni). (**D**) Cumulative distribution plot of length of novel repeat elements within the orthologous downstream introns of conserved, species-specific and background circRNAs from lemur to human (see *Figure 2D* for description of cumulative distribution plots). p-Values calculated by Wilcoxon rank-sum test and corrected for multitesting (Bonferroni). (**E**) Pyramid plot of the proportion of repeat elements inserted into the downstream introns of conserved, species-specific and background circRNAs from lemur to human. *p<0.05; **p<0.005, ***p<1 × 10⁻⁵. p-Values calculated by Wilcoxon rank-sum test and corrected for multitesting (Bonferroni). (**F**) A schematic model of the results describing impact of our observations on circRNA formation. Boxes represent exons, straight lines are introns, repeat elements are red, arced lines represent back-spliced

*Figure 4 continued on next page*

*Figure 4 continued*

junction, and dashed lines represent RNA-RNA duplex.

The online version of this article includes the following figure supplement(s) for figure 4:

**Figure supplement 1.** Cumulative distribution plot of length of introns adjacent to conserved circRNAs.

as evidence of recent transposons insertion and extended adjacent introns (*Ashwal-Fluss et al., 2014*; *Ivanov et al., 2015*; *Jeck et al., 2013*; *Liang et al., 2017*). In line with previous work, the majority of circRNAs identified arose from the same gene locus (alternative circularization) (*Burd et al., 2010*; *Jeck et al., 2013*; *Salzman et al., 2012*; *Zhang et al., 2014*); however, we identify that this phenomenon is largely limited to species-specific circRNAs and disappears in the conserved group. Similarly, we identify that the adjacent introns of circRNAs are significantly longer with inverted Alu repeats (*Ashwal-Fluss et al., 2014*; *Ivanov et al., 2015*; *Jeck et al., 2013*; *Liang et al., 2017*); however, only in the conserved group do we observe a bias towards lengthening of the downstream adjacent intron with inverted L1 repeats dominating. Finally, in contrast to previous work, we do not identify that conserved circRNAs are more strongly expressed but instead that conserved circRNAs have greater relative expression compared to linear transcript with this ratio increasing with the evolutionary age of the circRNA. This decreased diversity of conserved circRNA production and increased relative expression is in line with data from linear splicing (*Baek and Green, 2005*; *Barbosa-Morais et al., 2012*; *Gueroussov et al., 2017*; *Irimia et al., 2009*; *Merkin et al., 2012*) and suggests circRNA selection is occurring. However, an important limitation of our approach is our usage of annotated splice sites, thus limiting our conclusions to exonic circRNAs from canonical splice sites.

A host of endogenous mechanisms dampen down the impact of the retrotransposons within gene bodies. For example, the formation of Alu exons is suppressed by the nuclear ribonucleoprotein HNRNPC (*Zarnack et al., 2013*) and the nuclear helicase DHX9 binds to inverted repeat Alu elements to suppress circRNA formation (*Aktaş et al., 2017*). Over time though, in selected examples, these inclusions can promote novel functionality (*Attig et al., 2016*; *Attig et al., 2018*; *Avgan et al., 2019*; *Shen et al., 2011*), enabling the creation of tissue-specific exons (*Attig et al., 2018*), miRNAs (*Gu et al., 2009*; *Spengler et al., 2014*), and promoter regions (*Li et al., 2018a*; *Zhang et al., 2019*). Our results suggest that circRNAs are undergoing a similar selection race with the recent insertion of multiple retrotransposons promoting increased circRNA production that in some cases stabilizes over time. It is important to note though that the production of a large number of circRNAs in itself can be functional (*Liu et al., 2019*). For example, in the immune system a wide diversity of circRNAs are produced to sequester-specific RNA-binding proteins. These proteins are released upon viral infection to inhibit translation of viral RNA (*Liu et al., 2019*). A major challenge for the field in the following years will arise from determining the contribution of noise versus function for each of these groups.

The investigation of mechanisms controlling circRNA production is a rapid and expanding field (*Li et al., 2018b*). Our results support a kinetic model (*Schor et al., 2013*) for circRNA function whereby trans-factors promote spliceosome recruitment to the final exon and the very long downstream introns extend the time window for back-splicing to occur. This is facilitated by inverted repeats increasing the proximity of 3'-splice site with the upstream 5'-splice site (see *Figure 4G*). The extension of the final intron therefore increases the likelihood of circRNA formation in time and space. Spatially by introducing new retrotransposons, which facilitates RNA-RNA duplex formation (*Ivanov et al., 2015*; *Jeck et al., 2013*; *Li et al., 2017*; *Liang and Wilusz, 2014*) to orientate the splice sites in close proximity and temporary by increasing the intron length, it expands the time window for such an event to occur (*Veloso et al., 2014*), which acts independent of the rate of RNA polymerase II across the gene body (*Zhang et al., 2016*). This model conforms with the previous observations of enrichment of inverted repeat Alu elements and of long introns surrounding circRNAs (*Ashwal-Fluss et al., 2014*; *Dong et al., 2017*; *Ivanov et al., 2015*; *Jeck et al., 2013*; *Liang and Wilusz, 2014*; *Rybak-Wolf et al., 2015*; *Zhang et al., 2014*).

The conservation of circRNAs we observe could therefore just be a result of increasing the probability for such an event to occur rather than evidence of functionality. However, circRNAs represent an extreme example of a trend in post-transcriptional regulation whereby low leaky expression creates a pool of possible novel substrates (*Avgan et al., 2019*; *Barbosa-Morais et al., 2012*; *Fiszbein et al., 2019*; *Mattick, 2018*; *Merkin et al., 2012*; *Reyes et al., 2013*), increasing the likelihood for unique

functionality to arise (*Gueroussov et al., 2017*; *Guo et al., 2020*). For circRNAs, this can be aided by single-nucleotide changes that enable trans-acting factors, such as Quaking, ADAR, or NF90/110, to facilitate circRNA formation (*Conn et al., 2015*; *Ivanov et al., 2015*; *Li et al., 2017*).

In conclusion, our evolutionary analysis identifies that the noisy production of circRNAs is driven by the insertion of novel transposons in adjacent downstream introns that can over time stabilize to produce conserved circRNAs. This provides a pool of evolutionary potential that could contribute to the evolutionary rewiring of the cell.

## Materials and methods
### Data processing
All fastq files were quality-checked using FastQC (*Andrews, 2010*). Adapters and low-quality sequences were removed using Cutadapt (*Martin, 2011*).

### Datasets
Ribo-minus RNA-seq data was extracted from the publicly available Nonhuman Primate Reference Transcriptome Resource (NHPRTR) resource (http://www.nhprtr.org/; *Peng et al., 2015*). The analyzed samples were from chimpanzee, rhesus macaque, cynomolgus macaque mauritian, olive baboon, common marmoset, squirrel monkey, and mouse lemur to cover the ~70 millions of years (MYA) of primate evolution (*Supplementary file 1*). The primates samples of the above species were chosen based on the availability of chain files for LiftOver analysis. Human samples were retrieved from different publicly available *Ribo-minus* datasets searching for the SRA IDs in the circAtlas 2.0 database (http://circatlas.biols.ac.cn/; *Wu et al., 2020*; *Supplementary file 1*). Replicates of certain samples across the different primates data were merged to achieve a higher sequencing depth required for alternative splicing quantification (*Supplementary file 5*).

### Alternative splicing, back-splice junction, and gene expression quantification
Whippet (*Sterne-Weiler et al., 2018*) was used to analyze the RNA-seq samples to quantify cassette exon (CE) events, circRNAs (BSJs), and gene expression. To enable BSJ quantification, we used the setting with the `--circ` parameter when running Whippet-quant (https://github.com/timbitz/Whippet.jl, *Timothy, 2021*).

The splice graphs of all primates used for Whippet quantification were calculated using the genome annotation files for each primate from Ensembl (*Yates et al., 2020*; *Supplementary file 6*). The genome annotation files were supplemented with novel EEJs derived from whole-genome alignment of primates samples using STAR (*Dobin et al., 2013*) with the *2-pass setting* and *outFilterMultimapNmax==10* parameters. Whippet index command was run with the *--bam* and *--suppress-low-tsl* parameters.

Gene expression of orthologue genes was retrieved from the *gene.tpm.gz* files from Whippet-quant output. The correlations of gene expression of orthologue genes between tissue samples from all primates were calculated using Pearson's correlation. Clustering of correlation values was assessed and visualized with a heatmap using the p.heatmap function in R.

### Identification of expressed circRNAs and CEs
All the BSJ events present in orthologue genes between the species mentioned above were filtered to find conserved circRNAs identified by Whippet. The orthologue list of genes was retrieved from Ensembl using the bioMart R package (*Smedley et al., 2009*). Expressed BSJs were defined according to an expression and PSI cutoff of at least five reads and ≥5% of PSI, respectively. CE events from Whippet output were also filtered, keeping those present in orthologue genes and with PSI ≥ 10% .

### Conservation analysis of circRNAs
We defined a circRNA as conserved if the exon(s) that formed the BSJ are orthologous to the human exon(s) that also formed the BSJ. To achieve this, the exon coordinates of orthologue genes of each primate were retrieved from the GTF files downloaded from Ensembl (*Supplementary file 6*). Then, the exon coordinates from the GTF files were intersected with the CE coordinates from Whippet using bedtools intersect (*Quinlan and Hall, 2010*) with -wa parameter.

Then, the resulted exon coordinates (GTF-CE coordinates) were intersected with the circRNAs coordinates within orthologue genes using *bedtools intersect* with -*loj* parameter to find which exons were forming the circRNA. The exon coordinates within the circRNA coordinate of the non-human primates were mapped to human coordinates using the UCSC LiftOver (*Navarro Gonzalez et al., 2021*) to retrieve orthologue exons.

The orthologue exons between primates and human were matched to human exon coordinates within the circRNAs coordinates in human to find conserved circRNAs. We defined if a circRNA was conserved between a primate and human if the exon(s) forming the BSJ of the circRNA were also conserved and if the exon(s) start and end coordinates were ≤100 nc from the start and end of the BSJ coordinate (see *Figure 2* and S5 for schematic). We defined as non-conserved circRNAs all the human circRNAs that do not have orthologue exons forming the BSJ of the circRNA with other primates.

## Conserved and tissue-conserved circRNAs

The list of orthologous circRNAs was plotted in an UpSet plot to visualize the intersection of circRNAs between primates species. We defined the set of conserved circRNAs as the circRNAs within the intersections between primates species where human, chimpanzee, and baboon always appeared.

The correlation of inclusion of conserved and tissue-conserved circRNAs between all samples was calculated using Pearson's correlation. Then correlation values were plotted in a heatmap using the p.heatmap function in R.

## Differential gene expression analysis and enrichment analysis of genes with conserved circRNAs

EdgeR (*Robinson et al., 2010*) library was used to perform the differential gene expression analysis between neuronal samples (brain, cerebellum, and frontal cortex) and non-neuronal samples (heart, skeletal muscle, liver, lung, spleen, and colon). This analysis showed 8817 differentially expressed genes according to a log fold change cutoff of $log_2(1.5)$ and FDR of 0.05.

There were 212 genes of the conserved circRNAs (total of 442 genes) in the set of differentially expressed genes. The enrichment of genes with conserved circRNAs was statistically tested with a hypergeometric test using the phyper function in R. The parameters were q = 212, m = 8,817, n = 11,278, k = 442, and lower.tail = FALSE.

## Conserved CEs in primates

All exon coordinates of orthologue genes from the GTF files and CE exon coordinates from Whippet were mapped to human coordinates using UCSC LiftOver (*Navarro Gonzalez et al., 2021*). The PSI values of orthologous exons in genes of conserved and tissue-conserved circRNAs were retrieved from all tissue samples of human, chimpanzee, and baboon and calculated Pearson's correlation values. The correlation values were plotted in a heatmap using the p.heatmap function.

Comparison of circRNAs expression and conservation circRNAs expression of conserved, tissue-conserved, and non-conserved circRNAs was calculated using relative transcripts per million (TpMs). Relative TpMs are the expression of circRNAs measured in TpMs. Relative TpMs were calculated as the proportion of gene expression measured in TpMs relative to the number of reads of the circRNA using the formula

$$Relative\ TpMs = \frac{circRNA\ Reads * Gene\ TpM}{Gene\ Reads}$$

where *circRNA Reads* refers to the number of reads in the BSJ/circRNA, *Gene TpM* refers to the TpM value of the gene with the exons of the circRNA, and *Gene Reads* refers to the number of reads of the gene with the exons of the circRNA.

The expression values of conserved and non-conserved circRNAs, and tissue-conserved and non-conserved circRNAs of replicates of the same tissue in human samples were plotted in scatter plots.

The median relative TpMs of conserved (and tissue-conserved) and non-conserved circRNAs of human samples were also calculated. The expression values between mentioned sets were statistically compared using a Wilcoxon test. The parameters of the Wilcoxon test were x = conserved (or tissue-conserved) circRNAs TpMs, y = non-conserved circRNAs TpMs, alternative = 'greater.' The median relative TpM was plotted in violin plots using the ggplot2 R library (*Wickham, 2016*).

The median PSI values of conserved, tissue-conserved, and non-conserved circRNAs across all human samples were calculated. Their inclusion levels were statistically compared using the Wilcoxon test function in R with the parameters x = conserved (or tissue-conserved) circRNAs median PSI, y = non-conserved circRNAs median PSI, alternative = 'greater.' The distribution of the median PSI values of conserved and non-conserved circRNAs, and tissue-conserved and non-conserved circRNAS was plotted in a cumulative plot using the ggplot2 library in R.

The median PSI values of shared circRNAs between evolutionary interesting sets (human [species-specific circRNAs]; hominoids; hominoids and baboon; hominoids and old-world monkeys; hominoids, old-world monkeys and marmoset; and hominoids, old-world monkeys and new-world monkeys) shown in the UpSet plot were calculated, plotted in a cumulative plot, and statistically compared using a Wilcoxon test.

Seven of our reported circRNAs from the lists of conserved and tissue-conserved circRNAs were of special interest as they were previously reported (*Gokool et al., 2020a*) to be highly expressed in human cerebellum and frontal cortex. The PSI values of such circRNAs were compared across all tissues in the eight primates species.

## Comparison of the number of orthologue genes producing a circRNA and number of conserved circRNAs between species

The number of times an orthologue gene produces at least one circRNA in any of the analyzed species was counted, as well as the number of times a circRNA was shared between another primate. The percentage of shared genes or circRNAs between the eight species was calculated and plotted in a barplot using the ggplot2 library in R.

Comparison of start and end position of circRNAs between conserved and non-conserved circRNAs circRNAs can be formed from unique start and end exons forming the BSJ, repeated start exons, repeated end exons, or repeated start and end exons (see *Figure 2—figure supplement 3* for schematic). The percentage of conserved and non-conserved circRNAs that fall in the above categories was calculated and plotted using the ggplot2 library in R.

## Generalized logistic regression

All continuous data were normalized to ensure a fair comparison between features using scale() package in R environment. Multicollinearity was assessed using the vif() from the R package *car*.

The dataset was split into training (80%) and test (20%). To optimize the selection of the model and the importance of each feature, we used the R package *glmulti* (*Calcagno and De Mazancourt, 2010*). To select from all possible models, the selection process used a genetic algorithm (method = 'g') with Akaike information criterion (AIC – crit = 'aic'). To calculate the generalized logistic model, glmulti used the R module *glm* with family = binomial(). ROC curve was calculated using R's *pROC* library with test data. Data extracted from this model is reported together with p-value and z-values in *Supplementary file 7*.

## Background datasets

Two background datasets were used in this study: background and species-specific (*Supplementary file 2*). The 'background' datasets consisted of exon combinations only within genes with circRNAs. The dataset was constructed by identifying alternative exons within gene of interest (10 < PSI < 90 l within any of the tissues studied) and using Python function *random* to assign these exons together. The 'species-specific' dataset was constructed as described above of human circRNA with no evidence of their BSJ being conserved in any other primate species. For both datasets, only genes with orthologous genes in all tested primates species were used (based on Ensembl annotation) and only orthologous exons (based on LiftOver – see above) were used.

## circRNA features

MaxEntScan (*Yeo and Burge, 2004*) was used to estimate the strength of 3' and 5'-splice sites. 5'-splice site strength was assessed using a sequence including 3 nt of the exon and 6 nt of the adjacent intron. 3'-splice site strength was assessed using a sequence including −20 nt of the flanking intron and 3 nt of the exon. SVM-BPfinder (*Corvelo et al., 2010*) was used to estimate branchpoint

and polyprimidine tract strength and other statistics. Scores were calculated using the sequence of introns to the 3′ end of exon between 20 and 500 nt.

Transcription start sites (TSS) were downloaded from Biomart. GC content was calculated using Python script. Transposon information was download from RepeatMasker as described below.

Nucleosome occupancy for HepG2 cells was calculated using data from *Enroth et al., 2014*. Color-space read data was aligned using Bowtie (*Langmead, 2010*) (-S -C -p 4 m 3 `--best` –strata) using index file constructed from Ensembl Hg38. Nuctools (with default settings) was used to calculate occupancy profiles and calculate occupancy at individual regions (*Vainshtein et al., 2017*).

All CLiP-seq data and CHiP-seq data were downloaded preprocessed bed data files from ENCODE (*Sundararaman et al., 2016*) with only narrowpeaks calculated using both isogenic replicates used. Bedtools intersect (-wao) was used to identify overlap with candidate regions. Overlap for all groups of trans-factors was collated and scores normalized by nucleotide length. Groups were based on annotation and split into positive regulators of splicing (SR: serine/arginine region containing proteins) and negative regulators of splicing (hnRNP: heterogeneous nuclear ribonucleoproteins).

In feature analysis, only first and last exons of circRNA, and their surrounding introns, were included in the analysis. The upstream portion is considered as the region 5′ of elements (i.e., first exon) and downstream portion is 3′ of elements.

## Overlap with known repeat elements

Repeat elements identified by RepeatMasker were downloaded from UCSC table browser (*Navarro Gonzalez et al., 2021*) in bed format. Bedtools intersect (−wao) was used to identify overlap of transposons with novel exons.

The frequency of transposable events is calculated as the proportion of transposons overlapping area of interest (i.e., exon 1). All transposons were grouped together into 12 categories (AluJ, AluS, AluY, L1, L2, L3, MIR, MER, FLAM, AT_rich, SINE, and everything else into 'other') based on annotation from RepeatMasker. Inverted repeat regions are defined as having the same transposable elements on different strands in both introns adjacent to the circRNA.

## Intronic length and transposons comparison of human and lemur

Orthologous exons between human and lemur containing circRNAs were identified using the procedure described above. Intron length was determined based on the nearest exon from Ensembl annotation (*Yates et al., 2020*) with evidence from RNA-seq data of expression (PSI > 10). To identify regions unique to human, the intronic regions unique to human were split into windows of 20 nt. LiftOver was used to identify conserved regions between human and lemur genomes for each of these windows. Regions with no evidence of conservation were overlapped (using bedtools intersect –wao) with UCSC RepeatMasker (*Navarro Gonzalez et al., 2021*) annotation to identify novel transposon insertion.

Previously reported circRNAs from circAtlas circRNAs reported in the circAtlas database were downloaded from their webserver (http://circatlas.biols.ac.cn/). As the circRNA coordinates in the bed file had all types of circRNAs, we used bedintersect to keep only those circRNAs from annotated exons (hg38 GENCODE). Using bedtools, the filtered exonic circRNAs from circAtlas were intersected with the conserved and species-specific circRNAs to calculate the percentage of shared circRNAs.

## Benchmarking Whippet for circRNA detection

Whippet has been previously benchmarked for the detection of linear splicing events (*Sterne-Weiler et al., 2018*). However, it has not been previously validated for detection of back-splicing events that create circRNAs. To benchmark Whippet's performance on circRNA detection, we analyzed both circRNA detection and computational performance.

## Simulated dataset comparison

CIRIsimulator (*Gao et al., 2015*) was used to make four simulated datasets with sequencing levels of 10-, 20-, 30-, and 40-fold read depth. Simulated sequencing data was generated using the chromosome 1 fasta from the hg19 human genome and its GTF annotation file obtained from the CIRI software repository (https://sourceforge.net/projects/ciri/). The parameters used were default insert length, 75 read length, and no sequencing errors.

With the simulated datasets, we ran Whippet (*Sterne-Weiler et al., 2018*), CIRCexplorer3 (*Ma et al., 2019*), CIRIquant (*Zhang et al., 2020*), and find_circ (*Memczak et al., 2013*). Whippet parameters were the same as previously described (see Materials and methods). CIRCexplorer3 was run using CIRCexplorer2 output file (https://github.com/YangLab/CLEAR, *Xiao-Ou, 2021*). To run CIRCexplorer2, we used the 'run with One Command' option of CIRCexplorer2 (https://circexplorer2.readthedocs.io/en/latest/tutorial/one_step/). In line with recommendation from authors, we used STAR to map the RNA-seq reads according to defined parameters (https://circexplorer2.readthedocs.io/en/latest/tutorial/alignment/). CIRIquant and find_circ were run according to the recommended parameters for each program (*Ma et al., 2019*; *Memczak et al., 2013*; *Sterne-Weiler et al., 2018*; *Zhang et al., 2020*). The performance of the programs was evaluated by assessing the number of circRNAs found versus the number of circRNAs in the simulated datasets.

## RNase R samples analysis

RNase R samples from human and macaque were downloaded from SRA database after defining a curated list of potential samples to analyze. Info about SRA ID, the title of the sample, and sequencing depth is given in *Supplementary file 1*.

The quality of samples was analyzed with FastQC (*Andrews, 2010*) and, if needed, adapters and low-quality sequences were trimmed using Cutadapt (*Martin, 2011*).

Quantification of circRNAs using Whippet was done as previously described for each corresponding primate. In the case of human samples, for the set of species-specific circRNAs, there was 62.3% of overlap (*Figure 1—figure supplement 2B*).

## Macaque RNase R samples analysis

As the set of conserved circRNAs is defined as 'all circRNAs present at least in human, chimpanzee, and baboon,' we first filter all those conserved circRNAs that are present in the macaque samples. According to this filter, we found 454 conserved circRNAs also conserved in macaque (conserved-macaque circRNAs). From the total of conserved-macaque circRNAs, we calculated the percentage of shared conserved-macaque circRNAs in the RNase R dataset. circRNAs in the RNase R dataset were defined as expressed with a $\geq 2$ reads cutoff.

Conserved-macaque circRNAs were also filtered to keep those with neuronal tissue expression. Neuronal tissue expression of the circRNAs was defined as all those circRNAs that had a PSI value (in neuronal samples: cerebellum and frontal cortex samples) of at least 5% . From this filter, there are 385 conserved-macaque circRNAs with neuronal expression. The percentage of shared of circRNAs with the RNase dataset was also calculated. The circRNAs in the RNase R dataset was defined as expressed with a $\geq 2$ reads cutoff.

## False-positive rate

PolyA+ and ribodepleted strand RNA-seq data from human brain regions samples (*Gokool et al., 2020a*; *Supplementary file 1*) were analyzed with Whippet, CIRCexplorer3, CIRIquant, and find_circ using the recommended parameters for each program (*Ma et al., 2019*; *Memczak et al., 2013*; *Sterne-Weiler et al., 2018*; *Zhang et al., 2020*). Indices needed for mapping reads were built using the hg38 genome version and with default parameters. All circRNAs from all the programs were defined to be expressed with a $\geq 5$ reads cutoff. The false-positive rate for each program was calculated as the percentage of circRNAs shared between polyA+ and ribodepleted samples. We calculated the FPR of Whippet, CIRCexplorer3, CIRIquant, and find_circ. The false-positive rate was calculated as the percentage of circRNAs shared between polyA+ and ribodepleted samples with previous reports showing FPR < 2 (*Gokool et al., 2020b*) and with other reports finding that polyA+-based FPR of many algorithms ranges from ~3% to 8% (*Szabo et al., 2015*).

## Time and memory computation comparison

Quantification of time and memory used for each of the programs (Whippet [*Sterne-Weiler et al., 2018*], CIRCexplorer3 [*Ma et al., 2019*], CIRIquant [*Zhang et al., 2020*], and find_circ [*Memczak et al., 2013*]) was done using the built-in time function in GNU Linux, version 1.7, when analyzing the same sample (GOK5490A11_S15_ba9RD) from the ribodepleted dataset. The total run time was calculated as the sum of user time and system time from the time program output. The memory used

for each program is the maximum resident set size value from the time program output. Time and memory quantification was done for each of the steps needed to get the final output of the circRNA quantification without considering building indices for each mapping program. Total time was transformed from seconds to minutes and total memory from kbytes to Gbytes.

### Gene expression of genes with conserved and species-specific circRNAs

Gene expression of genes with exons from conserved circRNAs was compared with the gene expression of genes with exons from species-specific circRNAs. The gene expression comparison was done in each tissue and the median expression of all tissue samples (*Supplementary file 1*). In the case of each tissue comparison, the mean gene expression (TpM) was calculated for all replicates of each tissue.

In all the gene expression comparisons (tissue-specific and median tissue expression), the set of gene expression of conserved circRNAs was statistically compared with the set of gene expression of species-specific circRNAs using the Wilcoxon rank-sum test. Gene expression distribution of both sets of genes was transformed to log2 and then plotted in violin plots.

### Comparison of number of exons between conserved and species-specific circRNAs

The number of exons in conserved and species-specific circRNAs was quantified according to the number of exons that were present in the BSJ of the circRNAs. The exon coordinates were defined according to Ensembl and all exons most have evidence of expression ($\geq 5$ reads and $\geq 5\%$ PSI). The distribution of the number of exons was plotted in violin plots and statistically tested using Wilcoxon rank-sum test in R with the parameter alternative = 'less'. To test if conserved circRNAs were enriched in circRNAs species with number of exons of 2–3, we performed Fisher's exact test in R with the parameter alternative = 'greater.' For this analysis, we defined the below contingency table:

|  | CircRNAs with 2–3 exons | CircRNAs with more or with less of 2–3 exons |
|---|---|---|
| Conserved | 198 | 575 |
| Species-specific | 1966 | 9235 |

## Acknowledgements

We gratefully acknowledge John Mattick, Akira Gookol, Juli Wang, and Helen King for helpful discussions and feedback on this study. GSR was supported by a UNSW PhD Scholarship. This research was supported by the NSW Institute of Cancer Research (RJW), the Scrimshaw Foundation (RJW), the Australian Research Council (ARC) Discovery Project (RJW, IV), an ARC future fellowship (IV), and a University of New South Wales Scientia Fellowship (IV).

## Additional information

### Funding

| Funder | Grant reference number | Author |
|---|---|---|
| Australian Research Council |  | Irina Voineagu<br>Robert J Weatheritt |
| Cancer Institute NSW |  | Robert J Weatheritt |
| UNSW Australia | PhD Scholarship | Gabriela Santos-Rodriguez |
| University of New South Wales | Scientia Fellowship | Irina Voineagu |
| Nutrition Society | Scrimshaw Foundation | Robert J Weatheritt |
| Australian Research Council | (ARC) Discovery Project | Robert J Weatheritt<br>Irina Voineagu |

| Funder | Grant reference number | Author |
|---|---|---|

The funders had no role in study design, data collection and interpretation, or the decision to submit the work for publication.

## Author contributions
Gabriela Santos-Rodriguez, Robert J Weatheritt, Conceptualization, Formal analysis, Funding acquisition, Investigation, Methodology, Project administration, Resources, Software, Supervision, Validation, Visualization, Writing – original draft, Writing – review and editing; Irina Voineagu, Conceptualization, Funding acquisition, Writing – review and editing

## Author ORCIDs
Gabriela Santos-Rodriguez ![ORCID] http://orcid.org/0000-0002-9076-0294
Robert J Weatheritt ![ORCID] http://orcid.org/0000-0003-3716-1783

## Decision letter and Author response
Decision letter https://doi.org/10.7554/eLife.69148.sa1
Author response https://doi.org/10.7554/eLife.69148.sa2

## Additional files

### Supplementary files
- Supplementary file 1. Dataset IDs and sequencing depth information.
- Supplementary file 2. Non-conserved (human-specific) and background circular RNAs.
- Supplementary file 3. Associated with circular RNA formation (trans- and cis-regulatory factors and all major groups of transposons).
- Supplementary file 4. Specific circular RNAs.
- Supplementary file 5. About merged samples to acquire higher sequencing depth.
- Supplementary file 6. Version, GTF, and chain file information.
- Supplementary file 7. Output.
- Transparent reporting form

### Data availability
All datasets used in this study are included in the manuscript and supporting files. Analyzed data is also included in supporting material as well.

The following previously published datasets were used:

| Author(s) | Year | Dataset title | Dataset URL | Database and Identifier |
|---|---|---|---|---|
| Peng X, Thierry-Mieg J, Thierry-Mieg D, Nishida A, Pipes L, Bozinoski M, Thomas MJ, Kelly S, Weiss JM, Raveendran M, Muzny D, Gibbs RA, Rogers J, Schroth GP, Katze MG, Mason CE | 2015 | Tissue-specific RNA-sequencing for ten non-human primate species | https://www.ebi.ac.uk/ena/browser/view/PRJNA271912 | ENA, PRJNA271912 |
| Nielsen MM, Tehler D, Vang S, Sudzina F, Hedegaard J, Nordentoft I, Orntoft TF, Lund AH, Pedersen JS | 2014 | Identification of expressed and conserved human non-coding RNAs | https://www.ebi.ac.uk/ena/browser/view/PRJNA193501 | ENA, PRJNA193501 |

*Continued on next page*

*Continued*

| Author(s) | Year | Dataset title | Dataset URL | Database and Identifier |
|---|---|---|---|---|
| Zheng Q, Bao C, Guo W, Li S, Chen J, Chen B, Luo Y, Lyu D, Li Y, Shi G, Liang L, Gu J, He X, Huang S | 2016 | RNA Sequencing Facilitates Quantitative Analysis of Transcriptomes in Human Normal and Cancerous Tissues | https://www.ebi.ac.uk/ena/browser/view/PRJNA311161 | ENA, PRJNA311161 |
| Bush SJ, McCulloch MEB, Summers KM, Hume DA, Clark EL | 2017 | Production ENCODE transcriptome data | https://www.ebi.ac.uk/ena/browser/view/PRJNA30709 | ENA, PRJNA30709 |
| Zhang Y, Zhang XO, Chen T, Xiang JF, Yin QF, Xing YH, Zhu S, Yang L, Chen LL | 2013 | RNA-seq of RNase R treated poly(A)-/ribo- RNAs from H9 cells | https://www.ebi.ac.uk/ena/browser/view/PRJNA208625?show=reads | ENA, PRJNA208625 |
| Panda AC, De S, Grammatikakis I, Munk R, Yang X, Piao Y, Dudekula DB, Abdelmohsen K, Gorospe M | 2017 | High-purity circular RNA isolation method (RPAD) reveals vast collection of intronic circRNAs (IcircRNAs) | https://www.ebi.ac.uk/ena/browser/view/PRJNA358203?show=reads | ENA, PRJNA358203 |
| Chen S, Huang V, Xu X, Livingstone J, Soares F, Jeon J, Zeng Y, Hua JT, Petricca J, Guo H, Wang M, Yousif F, Zhang Y, Donmez N, Ahmed M, Volik S, Lapuk A, Chua MLK, Heisler LE, Foucal A, Fox NS, Fraser M, Bhandari V, Shiah YJ, Guan J, Li J, Orain M, Picard V, Hovington H, Bergeron A, Lacombe L, Fradet Y, Têtu B, Liu S, Feng F, Wu X, Shao YW, Komor MA, Sahinalp C, Collins C, Hoogstrate Y, de Jong M, Fijneman RJA, Fei T, Jenster G, van der Kwast T, Bristow RG, Boutros PC | 2019 | RNA-Seq with and without RNase treatment in PCa cell lines | https://www.ebi.ac.uk/ena/browser/view/PRJNA450077 | ENA, PRJNA450077 |
| Yaylak B, Erdogan I, Akgul B | 2019 | Transcriptomics analysis of circular RNAs differentially expressed in apoptotic HeLa cells | https://www.ebi.ac.uk/ena/browser/view/PRJNA515690?show=reads | ENA, PRJNA515690 |
| Xiao MS, Wilusz JE | 2019 | An improved method for circular RNA purification that efficiently removes linear RNAs containing G-quadruplexes or structured 3' ends | https://www.ebi.ac.uk/ena/browser/view/PRJNA541935?show=reads | ENA, PRJNA541935 |
| Mahmoudi E, Kiltschewskij D, Fitzsimmons C, Cairns MJ | 2019 | CircRNA analysis in depolarized neuroblastoma cells | https://www.ebi.ac.uk/ena/browser/view/PRJNA578068?show=reads | ENA, PRJNA578068 |

*Continued on next page*

*Continued*

| Author(s) | Year | Dataset title | Dataset URL | Database and Identifier |
|---|---|---|---|---|
| Conn VM, Gabryelska M, Marri S, Stringer BW, Ormsby RJ, Penn T, Poonnoose S, Kichenadasse G, Conn SJ | 2020 | Role of SRRM4 in regulating microexons in Circular RNAs | https://www.ebi.ac.uk/ena/browser/view/PRJNA625891?show=reads | ENA, PRJNA625891 |
| Conn VM, Gabryelska M, Marri S, Stringer BW, Ormsby RJ, Penn T, Poonnoose S, Kichenadasse G, Conn SJ | 2020 | Role of SRRM4 in regulating microexons in Circular RNAs in brain tissue | https://www.ebi.ac.uk/ena/browser/view/PRJNA668150?show=reads | ENA, PRJNA668150 |
| Haque S, Ames RM, Moore K, Lee BP, Jeffery N, Harries LW | 2020 | Islet-expressed circular RNAs are associated with type 2 diabetes status in human primary islets and in peripheral blood | https://www.ebi.ac.uk/ena/browser/view/PRJNA607015?show=reads | ENA, PRJNA607015 |
| Xu K, Chen D, Wang Z, Ma J, Zhou J, Chen N, Lv L, Zheng Y, Hu X, Zhang Y, Li J Li J | 2018 | Annotation and functional clustering of circRNA expression in rhesus macaque brain during aging | https://www.ebi.ac.uk/ena/browser/view/PRJNA363074?show=reads | ENA, PRJNA363074 |

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
