## [Decision Letter]

**Acceptance summary:**

This manuscript provides an evolutionary perspective of tissue-specific circRNA expression across 70 million years of primate evolution. The authors find that most circRNAs are not conserved, with the exception of a subset of approx. 700 brain-specific circRNAs. Interestingly, those circRNAs are characterised by increased length of the downstream intron during evolution due to the recent insertion of transposons.

**Decision letter after peer review:**

Thank you for submitting your article "Evolutionary dynamics of circular RNAs in primates" for consideration by *eLife*. Your article has been reviewed by 2 peer reviewers, and the evaluation has been overseen by a Reviewing Editor and Patricia Wittkopp as the Senior Editor. The reviewers have opted to remain anonymous.

The authors analyzed circRNA expression in several tissues across 9 species representing 70 million years of primate evolution. They report that while most circRNAs are species-specific, with the exception of a subset of approx. brain-specific 700 circRNAs. Common features of these circRNAs are their high PSI values, low number of exons, unique back-splicing junction sites and, most significantly, increased lengths of their downstream introns, which is associated with the insertion of young transposons. The authors propose that increased intron length provides a longer kinetic window for back-splicing events, which favors the production of circRNAs.

These results are timely and interesting, given current controversies about the general functionality of circRNAs and our limited knowledge of their evolution and regulatory mechanisms. However in our opinion the manuscript is in need of substantial revisions in two main fronts:

a) Improved data analyses, including validation of the nature of the circRNAs detected by their pipeline (e.g. using RNaseR), filtering by gene expression, more stringent specificity controls and cross-validation by other dedicated software tools.

b) Further contextualization of the findings, including properly citing and discussing previous essential literature that reported related findings and conclusions and revising causality claims based upon correlative associations.

Essential revisions:

a) Improved data analyses,

a.1. It is essential to document some type of validation of the circRNAs detceted (e.g. by RNaseR), as many back-splicing junctions can be caused by artifacts of reverse transcription or alignment.

a.2. Filtering for gene expression. When comparing features of conserved circRNAs and species-specific circRNAs, it is important to use similarly-expressed circRNAs for comparison, as using highly-expressed conserved circRNAs to compare with relatively lowly-expressed species-specific circRNAs might lead to biased conclusions. Similarly-expressed groups of conserved circRNAs and species-specific circRNAs should be selected for analyses (Figure 3B). In the current version, the authors selected circRNAs with at least 5 reads and ≥ 5% PSI for their analysis. However, since the sequence depths of different species vary largely (supplemental table 1) from less than 20 millions of reads in some samples to more than 120 millions of reads in some other samples, this selection of circRNAs with at least 5 reads and ≥ 5% PSI is biased across species and would be problematic for cross sample comparison due to unbalanced sequencing depths.

a.3. The authors used Whippet to analyze both circRNAs and linear RNAs. However, this tool was originally developed to analyze canonical splicing events, and it is unclear how it performs on circRNA prediction. In addition, it has been suggested to combine some other tools, such as CIRCexplorer and MapSplice, together for reliable circRNA prediction (e.g. Hansen et al., Nucleic Acids Res 2016). Otherwise, the authors need to show convincingly that using Whippet alone is better than the suggested tools. In addition, the authors should carefully quantify circRNAs by considering different sequence depths across samples.

a.4. A number of publications have shown the association of long introns with circRNA expression, and more importantly, that the pairing between flanking introns of circRNAs is required for circRNA biogenesis in mouse and human. In line 42, an important reference, Zhang et al., Cell 2014, PMID: 25242744, should be cited to mention that "inverted repeat elements that promote complementarity between adjacent introns favouring circRNA formation". Similar conclusions may be reached in this study. For example, in Figure 3B, L2_flank and L1_flank can be the key features that discriminate the conserved and species-specific circRNAs, while the pairing between them in flanking introns may be more important than intron lengths for circRNA formation. In this case, it might be biased to emphasize the importance of downstream intron length for evolutionary circRNA biogenesis.

a.5. To show the significance of the downstream intron lengths for conserved circRNA formation, the authors should use lengths of upstream introns of conserved circRNAs as internal controls.

a.6. It is also not clear how to solve the problem caused by multicollinearity when predicting elements for circRNA biogenesis. It is possible a collinearity between long downstream introns with their pairing between flanking (long) upstream and downstream introns.

a.7. It has been shown that expressed circRNAs are enriched with 2-3 exons in multiple early studies. In this case, the low number of exons might not be specific for conserved circRNAs, but commonly among expressed circRNAs. The authors can compare the numbers of exons of conserved circRNAs with those of expressed (except conserved circRNAs) and all expressed circRNAs.

a.8. On page 18 the authors defined relative TPM values. However, the definitions of circRNAsRead and GeneRead are not clear.

b) Further contextualization of the findings. Relevant aspects of the authors' findings have been previously reported and should be properly cited and discussed, including

b.1. The reports by Rybak et al. (Mol Cell 2015), Veno et al. (Genome Biol. 2015) and You et al. (Nature Neuro 2015, which is not even cited) described that brain circRNAs are conserved, as well as several of the features described for these circRNAs in the manuscript.

b.2. Consider the orientation of the inserted transposons (as showed in Chen LL et al., Cell 2014)

b.3. Extension of downstream introns and inefficient cleavage and polyadenylation are well described as factors modulating exon circularization (see Liang et al. Mol Cell 2017, Ashwall et al., Mol Cell 2014).

b.4. In the abstract the authors state that "many primate genes produce non-coding circular RNAs". This statement is not precise as it is not clear how many of the circRNAs are coding and how many non-coding. The non-coding should be eliminated. Moreover, seems like the authors don't even know about the research showing that some circRNAS are translated.

b.5. In line 41, the authors stated that "Back-splicing occurs co-transcriptionally" but it has been shown that back-splicing might occur not only co-transcriptionally (Ashwall Fluss et al. Mol Cell 2014) but both co-transcriptionally and post-transcriptionally (Zhang et al., Cell Rep 2016, PMID: 27068474).

b.6. The most important model of this paper is that insertion of young transposon into downstream introns results in longer intronic regions, which delay the RNA polymerase II to the next splice site and increase the possibility of circRNA conservation. However, it has been reported that "Pol II accelerates dramatically while transcribing through genes, but slows at exons" (Jonkers et al., *eLife* 2014) and "back-splicing outcomes correlate with fast RNA Polymerase II elongation rate" (Zhang et al., Mol Cell 2016). In this case, longer introns might lead to fast, but not slow RNA polymerase II for circRNAs.

b.7. This manuscript mainly showed the insertion of LINE for circRNA expression and conservation. However, previous studies have suggested the importance of SINE elements, especially Alu elements of primates, in circRNA biogenesis (for example, Jeck et al., RNA 2013, PMID: 23249747; Zhang et al., Cell 2014, PMID: 25242744; Dong et al., RNA Biol 2017, PMID: 27982734). Did the authors observe that Alu is less involved in primate circRNA expression than LINE? Or, since Alu is prevalent among primates, their contribution to conserved circRNAs is less important than LINE?

b.8. In line 39, the authors cited Guo et al., Cell 2020 to show as examples of functional circular RNAs especially in the immune and nervous systems, but it has nothing to do with circular RNAs.

b.9. In line 44, Quaking was not likely to be suggested to facilitate "these (assumedly inverted repeat elements) RNA-RNA interactions", instead, ADAR (Ivanov et al., Cell Rep 2015, PMID: 25558066; Rybak-Wolf et al., Mol Cell 2015, PMID: 25921068), DHX9 (Aktaş et al., Nature 2017, PMID: 28355180), NF90/110 (Li et al., Mol Cell 2017, PMID: 28625552) were suggested to be involved in inverted repeat element RNA-RNA interactions to regulate circular RNA biogenesis.

b.10. In line 67, the authors cited Pipes et al., Nucleic Acids Res 2013 to suggest their used datasets in this study. However, many samples originally in Pipes et al., Nucleic Acids Res 2013 had only mRNA-seq, without total RNA-seq Total RNA-seq datasets for many those samples were updated in Peng et al., Nucleic Acids Res 2015, which should be referenced.

c) Revising causality claims based upon associative correlations:

c.1. In line 101 the authors state "Many orthologous genes consistently express circRNAs even of the precise back-spliced junction is not conserved implicating importance of trans-factors in controlling cirRNA formation". There is no logic in this statement, long introns and random insertion of repetitive elements in inverse orientation could explain this w/o invoking any trans-acting factor.

c.2. The authors claim that "Anaylsis of the exonic structure of conserved circRNAs, showed that conserved circRNAs contain fewer exons, and rarely overlap with other circRNAs (Figure 2G, p = 4:08. 10*64, Fisher exact test; see Methods) displaying back-splicing at unique 5´- and 3´-splice sites. This indicates that these conserved circRNAs possess unique cis- or trans-regulatory features that enable a tight control of the number of exons within a circRNA and the back-spliced junctions used." This is not necessarily like that and the 2 aspects could be concurrent/codependent on a different factor (i.e. intron length).

c.3. After looking for predictive genomic features for circRNA biosynthesis the authors conclude that: "This indicates a core set of 24 cis- and trans-regulatory features drive the conserved formation of circRNAs compared to our background set of introns" This is factually and conceptually wrong, as opredictive value does not mean causality. So likely most of these factors are co-occurring, which is still interesting but should not be overstated.

*Reviewer #2:*

In this manuscript entitled "Evolutionary dynamics of circular RNAs in primates", Gabriela Santos-Rodriguez et al. analyzed the circRNA expression in severl tissue samples across 9 primate species. They showed that most circRNAs are species-specifically expressed, while a subset of circRNAs were neural tissue-specifically expressed across species. These conserved circRNAs commonly had high PSI values, low number of exons and unique back-splicing junction sites. After evaluated hundreds of potential regulatory elements that might regulate circRNA biogenesis, the authors found that the insertion of young transposons, which expands the lengths of downstream introns, may up-regulate the expression of circRNAs and could be involved in conserved circRNA expression. Although obtained these interesting conclusions, there are key concerns related with their applied methods and claims. For example, the authors used Whippet to analyze both circRNAs and linear RNAs. However, this tool was originally developed to analyze canonical splicing events, and it is unclear how it performs on circRNA prediction. In addition, it has been suggested to combine some other tools, such as CIRCexplorer and MapSplice, together for reliable circRNA prediction (Hansen et al., Nucleic Acids Res 2016 and etc.).

The main and most important model of this paper is that insertion of young transposon into downstream introns results in longer intronic regions, which delay the RNA polymerase II to the next splice site and increase the possibility of circRNA conservation. However, it has been reported that "Pol II accelerates dramatically while transcribing through genes, but slows at exons" (Jonkers et al., *eLife* 2014) and "back-splicing outcomes correlate with fast RNA Polymerase II elongation rate" (Zhang et al., Mol Cell 2016). In this case, longer introns might lead to fast, but not slow RNA polymerase II for circRNAs.

Meanwhile, a number of published references have shown the association of long introns with circRNA expression, and more importantly, the pairing between flanking introns of circRNAs is required for circRNA biogenesis in mouse and human. Similar conclusion can be obtained in this study. For example, in Figure 3B, L2_flank and L1_flank can be the key features that discriminate the conserved and species-specific circRNAs, while the pairing between them in flanking introns may be more important than intron lengths for circRNA formation. In this case, it might be biased to emphasize the importance of downstream intron length for evolutionary circRNA biogenesis.

When cited references of Barbosa-Morais et al., Science 2012 and Merkin et al., Science 2012, the authors stated that "gene expression is highly conserved between the same tissues in different species". However, Barbosa-Morais et al. have clearly mentioned in their abstract that "Within 6 million years, the splicing profiles of physiologically equivalent organs diverged such that they are more strongly related to the identity of a species than they are to organ type". In addition, Merkin et al. also stated that "alternative splicing is well conserved in only a subset of tissues and is frequently lineage-specific". Together, these cited papers emphasized the species-specific alternative splicing, which can be extended to back-splicing, as back-splicing is a new type of alternative splicing. In this case, alternative splicing (should include back-splicing) can, at least partially, explain the heterogeneous expansion in complexity across evolution.

Comments for the authors:

Additional analyses should be performed to address the concerns listed in public review with stringent setup for prediction and comparison. In addition, many mis-cited references should be also corrected, including those listed below.

1. The authors need to use other suggested tools that are specific for circRNA prediction for their analyses. Otherwise, the authors need to show convincing results that using Whippet alone is better than the suggested tools. In addition, the authors should carefully quantify circRNAs by considering different sequence depths across samples.

2. To show the significance of the downstream intron lengths for conserved circRNA formation, the authors may want to use lengths of upstream introns of conserved circRNAs as internal controls to draw this conclusion.

3. When comparing features of conserved circRNAs and species-specific circRNAs, it is better to used similarly-expressed circRNAs for comparison. Using highly-expressed conserved circRNAs to compare with relatively lowly-expressed species-specific circRNAs might lead to biased conclusion.

4. This manuscript mainly showed the insertion of LINE for circRNA expression and conservation. However, previous studies have suggested the importance of SINE elements, especially Alu elements of primates, in circRNA biogenesis (for example, Jeck et al., RNA 2013, PMID: 23249747; Zhang et al., Cell 2014, PMID: 25242744; Dong et al., RNA Biol 2017, PMID: 27982734). Did the authors observe that Alu is less involved in primate circRNA expression than LINE? Or, since Alu is prevalent among primates, their contribution to conserved circRNAs is less important than LINE?

5. More stringent comparisons and controls are needed throughout the study. For example, similarly-expressed groups of conserved circRNAs and species-specific circRNAs should be selected for analyses (Figure 3B). In the current version, the authors selected circRNAs with at least 5 reads and ≥ 5% PSI for their analysis. However, since the sequence depths of different species vary largely (supplemental table 1) from less than 20 millions of reads in some samples to more than 120 millions of reads in some other samples, this selection of circRNAs with at least 5 reads and ≥ 5% PSI is biased across species and would be problematic for cross sample comparison due to unbalanced sequencing depths.

6. It is also not clear how to solve the problem caused by multicollinearity when predicting elements for circRNA biogenesis. It is possible a collinearity between long downstream introns with their pairing between flanking (long) upstream and downstream introns.

7. It has been shown that expressed circRNAs are enriched with 2-3 exons in multiple early studies. In this case, the low number of exons might not be specific for conserved circRNAs, but commonly among expressed circRNAs. The authors can compare the numbers of exons of conserved circRNAs with those of expressed except conserved circRNAs and all expressed circRNAs.

8. In line 39, the authors cited Guo et al., Cell 2020 to show as examples of functional circular RNAs especially in the immune and nervous systems, but it has nothing to do with circular RNAs. Other correct references should be cited here. Please cite those correct references.

9. In line 41, the authors stated that "Back-splicing occurs co-transcriptionally", but it has been shown that back-splicing might occur both co-transcriptionally and post-transcriptionally (Zhang et al., Cell Rep 2016, PMID: 27068474). Please correct.

10. In line 42, an important reference, Zhang et al., Cell 2014, PMID: 25242744, should be cited here to show "inverted repeat elements that promote complementarity between adjacent introns favouring circRNA formation".

11. In line 44, Quaking was not likely to be suggested to facilitate "these (assumedly inverted repeat elements) RNA-RNA interactions", instead, ADAR (Ivanov et al., Cell Rep 2015, PMID: 25558066; Rybak-Wolf et al., Mol Cell 2015, PMID: 25921068), DHX9 (Aktaş et al., Nature 2017, PMID: 28355180), NF90/110 (Li et al., Mol Cell 2017, PMID: 28625552) were suggested to be involved in inverted repeat element RNA-RNA interactions to regulate circular RNA biogenesis.

12. In line 67, the authors cited Pipes et al., Nucleic Acids Res 2013 to suggest their used datasets in this study. However, lots of samples originally in Pipes et al., Nucleic Acids Res 2013 had only mRNA-seq, without totally RNA-seq, and totally RNA-seq datasets for many those samples were updated in Peng et al., Nucleic Acids Res 2015. In this case, Peng et al., Nucleic Acids Res 2015 should be more appropriate to be cited for clarifying their used datasets.

13. At page 18, authors defined relative TPM values. However, the definitions of circRNAsRead and GeneRead are not clear.

*Reviewer #3:*

In the manuscript entitled "Evolutionary dynamics of circular RNAs in primates", Gabriela Santos Rodriguez et al., investigate the evolution of circRNAs in primates. This is indeed a very important and timely topic given the doubts about the functionality of circRNAs and the little we know about their evolution. Briefly, the authors compare tissue specific transcriptomes across 70 million years of primate evolution. They found that most circRNAs are not conserved with the exception of a subset of approx. 700 brain specific circRNAs. Interestingly, the authors found that those circRNAs are defined by an extended downstream intron that is lengthening during evolution due to the insertion of transposons. While the findings are interesting and the manuscript timely, it doesn't bring much new information and seems to ignore essential knowledge in the field.

– The main findings of the manuscript have been previously reported and seem to be ignored by the authors, that they didn't cite and/or discuss them. For example, the reports by Rybak et al. (Mol Cell 2015), Veno et al. (Genome Biol. 2015) and You et al. (Nature Neuro 2015, which is not even cited) described that brain circRNAs are conserved as well as several of the features described for them in the manuscript. A lot of the work described there has been described in these and other reports. So the manuscript doesn't bring novelty regarding to the conservation of brain specific circRNAs.

– Moreover, the data analysis seems a little superficial. First, the authors utilized total RNAseq, without any type of validation (e.g. RNAseR). Moreover, the authors don't even mention the fact that many backsplicing junctions are usually artifacts of reverse transcription or alignment. Moreover, the authors don't seem to apply any filter to gene expression of the circRNAs and some of the criteria utilized is not really justified. For example, very abundant circRNAs might have low PSI if the host gene is expressed at high levels.

– The authors ignore literature in the field regarding circRNA biogenesis. This is important as the authors should consider the orientation of the inserted transposons (as showed in Chen LL et al., Cell 2014) when postulating their model. Moreover, extension of downstream introns and inefficient cleavage and polyadenylation are well described as factors modulating exon circularization (see Liang et al. Mol Cell 2017, Ashwall et al., Mol Cell 2014). This is not minor, as suggest the authors don't know (or chose to ignore) major literature in the field that shows that some of the findings are not new/novel.

– There are a lot of logical/conceptual mistakes in the text in which correlations are stated as causal relationships. A few examples of them are listed below:

o In line 101 the authors say "Many orthologous genes consistently express circRNAs even of the precise back-spliced junction is not conserved implicating importance of trans-factors in controlling cirRNA formation". There is no logic in this statement, long introns and random insertion of repetitive elements in inverse orientation could explain this w/o invoking any trans-acting factor.

o The authors claim that "Anaylsis of the exonic structure of conserved circRNAs, showed that conserved circRNAs contain fewer exons, and rarely overlap with other circRNAs (Figure 2G, p = 4:08. 10*64, Fisher exact test; see Methods) displaying back-splicing at unique 5´- and 3´-splice sites. This indicates that these conserved circRNAs possess unique cis- or trans-regulatory features that enable a tight control of the number of exons within a circRNA and the back-spliced junctions used." This is not necessary like that and the 2 things could be concurrent/codependent on a different factor (i.e. intron length).

o After looking for predictive genomic features for circRNA biosynthesis the authors conclude that: "This indicates a core set of 24 cis- and trans-regulatory features drive the conserved formation of circRNAs compared to our background set of introns" This is factually and conceptually wrong, predictive value does not mean causality. So likely most of these factors are co-current. This is still interesting but the attempt of overstating is alarming.

---

## [Author Response]

Essential revisions:a) Improved data analyses,a.1. It is essential to document some type of validation of the circRNAs detceted (e.g. by RNaseR), as many back-splicing junctions can be caused by artifacts of reverse transcription or alignment.

We fully agree with the dangers of false positives due to the artefacts of reverse transcription and alignment. This was the reason we required that all detected circular RNAs in our analysis were supported by at least 5 reads. We also agree that an orthogonal validation from an additional source would further increase confidence in the circRNAs we detected, as well as the subsequent analysis. We undertook this in two major ways:

Firstly, we overlapped our analysis with circRNAs previously identified in the peer-reviewed databases, circAtlas. This analysis revealed that 99.5% of conserved circRNAs and 97.03% of species-specific circRNAs in our analysis have been previously identified. Given that circAtlas used a different alignment program this increases our confidence that our results are not false positives due to alignment issues. To describe this result, we have added the following text to the Results section:

“To validate the quality of our identified circRNAs, we initially overlapped our data with circRNAs previously reported in circAtlas (Wu et al. 2020). This analysis found that 99.5% of the conserved circRNAs and 97.03% of species-specific circRNAs have been previously reported.”

Secondly, we curated a list of published human RNase R datasets (see Supplementary File 1). These datasets primarily came from cell lines, in contrast to our samples that were all from tissue data. Despite the known tissue and cell-specificity of circRNAs, we were still able to validate in RNase R data 82.7% of the conserved circRNAs (Figure 1—figure supplement 2A) and 62.3% of the species-specific circRNAs (Figure 1—figure supplement 2B). To further validate our findings that we can identify conserved circRNAs, we expanded our analysis to RNase R data from macaque data. In this analysis, we were able to validate ~85% of the conserved circRNAs (Figure 1—figure supplement 2E). All these results have been added to Results section and Methods. With the following text added to the Results section:

“Additionally, we verified our circRNAs dataset using Rnase R data (see Methods for details). […] To validate the conservation of our neuronal circRNAs, we next analyzed Rnase R samples from different brain macaque regions. This analysis identified ~89% of the conserved circRNAs (324 conserved circRNAs) (Figure 1—figure supplement 2F, see Methods section for details).”

a.2. Filtering for gene expression. When comparing features of conserved circRNAs and species-specific circRNAs, it is important to use similarly-expressed circRNAs for comparison, as using highly-expressed conserved circRNAs to compare with relatively lowly-expressed species-specific circRNAs might lead to biased conclusions. Similarly-expressed groups of conserved circRNAs and species-specific circRNAs should be selected for analyses (Figure 3B). In the current version, the authors selected circRNAs with at least 5 reads and ≥ 5% PSI for their analysis. However, since the sequence depths of different species vary largely (supplemental table 1) from less than 20 millions of reads in some samples to more than 120 millions of reads in some other samples, this selection of circRNAs with at least 5 reads and ≥ 5% PSI is biased across species and would be problematic for cross sample comparison due to unbalanced sequencing depths.

We agree with the reviewer that differences in read depth of samples may bias to increased detection of circRNAs in datasets with greater depth. This would result in (i) the conserved circRNAs being within genes with higher expression than species-specific genes (ii) conserved circRNAs being more highly expressed.

We, therefore, compared the relative expression of genes with conserved circRNAs with genes with species-specific circRNAs (see Methods). This analysis showed no significant difference between the expression of genes with conserved circRNAs and genes with species-specific circRNAs (Figure 2—figure supplement 4B). To describe this result the following text has been added to the Results section of the manuscript:

“A comparison of genes containing conserved and species-specific circRNAs did not show any significant differences (Figure 2—figure supplement 4A and 4B, p = 0.584 Wilcoxon rank-sum test)”.

Despite, similar levels of gene expression the difference may still be due to tissue-specific gene expression of genes containing conserved circRNAs. We therefore assessed if the genes with conserved circRNAs had the same tissue-specificity as the conserved circRNAs (Figure 1—figure supplement 1B). Our results suggest that the conserved circRNA expression levels are not driven by the expression of the genes that contain the circRNA.

We have rewritten the Results section as follows to highlight this point more clearly:

“We next assessed if these changes may be explained by gene expression changes in the host gene. […] This suggests that circRNA conservation and expression is independent of these regulatory layers.”

We next assessed the expression of conserved circRNAs relative to species-specific circRNAs. If read depth were a major bias we would expect to observe that conserved circRNAs have an increased expression. However, our analysis reveals no significant differences of conserved versus species-specific circRNAs (Author response image 1. p< 0.187, Wilcoxon rank-sum test conserved vs species-specific; Figure 2C).

**Author response image 1. sa2fig1:** Conserved and species-specific circRNAs expression distributions. Violin plots describing relative expression levels of conserved and species-specific circRNAs. circRNA expression is measured in Relative TpM values. The difference in circRNA expression was statistically tested using the Wilcoxon rank-sum test.

We do agree though that it is important to highlight to the reader the differences in read depth. This is particularly an issue for Lemur. We have therefore added the proviso to the text associated with the UpSet plot in Figure 2—figure supplement 1.

The limited conservation of circRNAs to lemur may be influenced by low read depth of these samples (see Supplementary File 1).

Finally, we evaluated whether differences in the expression of the circRNAs resulted in changes in the features selected. To do this, we grouped our circRNAs by high, medium, and low expression bins. We then repeated our linear regression analysis on these groups. Remarkably, given that a decreased number of events will decrease statistical confidence in feature selection, we identify the top ten selected features are largely consistent across all three expression bins (Author response image 2).

**Author response image 2. sa2fig2:** Feature selection linear regression analysis according to circRNAs expression binsPlot showing the ranking of significance assigned by the linear regression model to features analyzed (y-axis) across the different groups of circRNA expression (high, medium and, low; x-axis).

a.3. The authors used Whippet to analyze both circRNAs and linear RNAs. However, this tool was originally developed to analyze canonical splicing events, and it is unclear how it performs on circRNA prediction. In addition, it has been suggested to combine some other tools, such as CIRCexplorer and MapSplice, together for reliable circRNA prediction (e.g. Hansen et al., Nucleic Acids Res 2016). Otherwise, the authors need to show convincingly that using Whippet alone is better than the suggested tools. In addition, the authors should carefully quantify circRNAs by considering different sequence depths across samples.

In this study, we have restricted circRNAs identification to only those using canonical and annotated splice sites, and to circRNAs derived from exons to avoid identification of circRNAs from cryptic splice sites, which is a major source of false positives in circRNA detection (Hansen et al. 2015). This is an important proviso to our analysis that we have now added to the Results section and we also mention it as a limitation in the Discussion section:

In the Results section: “For each species, we considered all primate-conserved internal exons as potential origins of back-spliced junctions with no restrictions on backward exon combination. Only canonical and annotated splice sites were used in the analysis.”

In the discussion: “An important limitation of our approach is our usage of annotated splice sites thus limiting our conclusions to exonic circRNAs from canonical splice sites.”

Whippet enables the detection of any combination of annotated exons. In the original paper, we only validated Whippet for combination of exons in the linear direction. The reviewer is correct that we did not benchmark Whippet for combination of exons in the reverse direction (i.e. circRNAs). To undertake this benchmark, we created a simulated dataset of circRNAs using the peer-reviewed program CIRI-simulator. This program stochastically selects whether a transcript will generate circular or linear reads. Combinations of exons to create the circRNAs are chosen at random. Reads for the linear RNAs and circRNAs are generated randomly from the RNA sequences. See the original article for more details (Gao et al. 2015. Genome Biology).

We used CIRI-simulator to generate different sequencing data from chromosome 1 using the chromosome 1 fasta from the hg19 human genome and its GTF annotation file obtained from the CIRI software repository (https://sourceforge.net/projects/ciri/). We selected read depths of 10, 20, and 40 fold with default insert length, read length of 75, and no sequencing errors (see Methods for details).

Whippet was run “—circ” flag to enable identification back-splicing and default settings (see methods). We compared Whippet to the most highly cited circRNA detection programs (CircExplorer3, find_circ, and CIRIquant). We used default recommended parameters as previously reported (see Methods for details). We benchmarked both the accuracy of all programs, plus their relative speeds and memory usage.

Whippet consistently identified ~90% of circRNAs simulated by CIRI-simulator across all read depths (Figure 1—figure supplement 3C). This suggests that read depth had relatively little impact on Whippet’s ability to detect circRNAs. At high read depths, CircExplorer 3 identified circRNAs with a similar detection rate to Whippet. However, CircExplorer v3 detection rate was more dramtically affected by read depth, as at lower read depths, its detection rate slipped to 77%. In contrast Whippet’s detection rate was 89.7%. Similarly for CIRIquant, at high read depths, it has a similar rate of detection than Whippet and CircExplorer 3. But, at low depths, the detection rate lowered to 82%. The sequencing depth most dramatically affects find_circ detection rate, where its lowest detection rate (42%) was found at the lowest sequencing depth and its highest detection rate (79%) was found at the highest sequencing depth (Figure 1—figure supplement 3C in Results section). Altogether, this suggests at high read depths Whippet and the majority of programs can successfully detect the majority of simulated circRNAs. However, in contrast to other programs, at lower read depths Whippet maintains its high detection rate.

We also analyzed matched datasets of poly(A) and ribo-minus RNA-seq data taken from the same samples to assess Whippet’s false positive rate (Gookol et al. 2019). This analysis revealed Whippet had a false positive rate (i.e. the percentage of circRNAs detected in DS1 data that were also detected in the polyA+ data from the same sample) of <2%, which is in line with FPR from other tools (Gookol et al. 2019).

We next benchmarked the speed and computational efficiency of the programs using the default Linux terminal program “time” (Methods). We used the sample GOK5490A11_S15_ba9RD from Gookol et al. 2019 that has a sequencing depth of almost 55M. Whippet directly analyses fastq RNA-seq files, and therefore comparisons to other algorithms included alignment steps, which used aligner recommended by each respective program. This analysis shows that Whippet analyses RNA-seq data for circRNAs in 1 hr 9 mins and 20s with a memory usage of just ~2 GB (Figure 1—figure supplement 3A and 3B). This is, on average, over 10x fold faster than other programs run with a considerably smaller memory overhead. Furthermore, Whippet also quantifies alternative splicing, alternative polyadenylation, and gene expression within this same timeframe.

We have added this information to the methods, as well as the following text to the Results section:

“The circRNA analysis was done using Whippet (Sterne-Weiler, Weatheritt, Best, Ha, & Blencowe, 2018)because, according to our benchmarking results (see Methods for details), it is an accurate and fast circRNA quantification tool. Our analysis of both simulated and collected RNA-seq data found that Whippet has a low false positive rate (< 2%, see Methods for details), which is in line with other methods (Szabo et al., 2015, Gookol et al., 2020), a high rate of circRNA identification even at low read depths (~90%; Figure 1—figure supplement 3C) and is faster (~69 minutes) with less computational overhead (< 3GB of memory on a single core) than other highly cited circRNA algorithms we compared with (circExplorer 3 (Ma et al., 2019), CIRIquant (Zhang et al., 2020), and find_circ (Memczak et al., 2013)) (Figure 1—figure supplement 3A and 3B).”

Together, this suggests Whippet is able to accurately detect circRNAs. Furthermore, Whippet is more robust to changes in read depth and is considerably quicker than all other tested programs.

a.4. A number of publications have shown the association of long introns with circRNA expression, and more importantly, that the pairing between flanking introns of circRNAs is required for circRNA biogenesis in mouse and human. In line 42, an important reference, Zhang et al., Cell 2014, PMID: 25242744, should be cited to mention that "inverted repeat elements that promote complementarity between adjacent introns favouring circRNA formation". Similar conclusions may be reached in this study. For example, in Figure 3B, L2_flank and L1_flank can be the key features that discriminate the conserved and species-specific circRNAs, while the pairing between them in flanking introns may be more important than intron lengths for circRNA formation. In this case, it might be biased to emphasize the importance of downstream intron length for evolutionary circRNA biogenesis.

The annotation “Flank” was a simplified annotation to reflect inverted repeat elements in the introns flanking the circRNAs, in line with the references cited by the reviewers. We acknowledge that this wording is unclear. We have therefore replaced the word “Flank” with “inverted repeat” throughout the manuscript and the figures in line with the literature.

In addition, we have rewritten the manuscript to more carefully reflect the important work in the literature.

In the introduction:

“Back-splicing occurs both co-and post-transcriptionally and is facilitated by inverted repeat elements that promote complementarity between adjacent introns favouring circRNA formation over linear splicing (Ivanov et al., 2015; Jeck et al., 2013; Liang and Wilusz, 2014; Zhang et al., 2014).”

In the results:

“This includes multiple features previously associated with conserved circRNAs, such as inverted-repeat Alu elements (Jeck et al., 2013; Zhang et al., 2014).”

In addition, we have refined Figure 3A, which now demonstrates the observation of Alu inverted repeats in all circRNAs versus background. We apologize for this confusion.

a.5. To show the significance of the downstream intron lengths for conserved circRNA formation, the authors should use lengths of upstream introns of conserved circRNAs as internal controls.

This is an excellent point. In our original paper, we used other introns within the same gene as an internal control but fully agree with the reviewer that the upstream intron of the conserved circRNA is an important control.

We, therefore, repeated our previous analysis using the upstream intron of the conserved circRNAs. Briefly, for each conserved circRNA in human, we identified using liftOver the orthologous lemur intron. This analysis revealed that the upstream intron on average remained stable across this evolutionary time period. In contrast, the downstream intron expanded significantly. This suggests that though both the upstream and downstream introns are longer than background introns, for our group of conserved circRNAs the expansion of the downstream intron correlates with the evolution of conserved circRNAs (Figure 4—figure supplement 1A).

In the results:

“In contrast to orthologous lemur introns, the human introns downstream of all identified circRNAs shows an almost four-fold expansion compared to the background dataset of introns within circRNA containing genes (Figure 4C, p < 3.84 x10^-23^ Wilcoxon rank-sum) and the upstream adjacent intron (Figure 4—figure supplement 1A, p < 1.02 x10^-10^ Wilcoxon rank-sum).”

a.6. It is also not clear how to solve the problem caused by multicollinearity when predicting elements for circRNA biogenesis. It is possible a collinearity between long downstream introns with their pairing between flanking (long) upstream and downstream introns.

Multicollinearity is an important control when undertaking linear regression. We assessed this using the *vif*() from the R package *car*. Please see methods section. However, we acknowledge it is important to clearly state in the Results section, as well. As such we have added the following line to the Results section:

“Using logistic regression combined with a genetic algorithm for model selection taking into account multicollinearity (see Methods)”.

In terms of upstream vs downstream introns, we refer review to our Reference A.5.

a.7. It has been shown that expressed circRNAs are enriched with 2-3 exons in multiple early studies. In this case, the low number of exons might not be specific for conserved circRNAs, but commonly among expressed circRNAs. The authors can compare the numbers of exons of conserved circRNAs with those of expressed (except conserved circRNAs) and all expressed circRNAs.

We thank the reviewer their observation. In the original manuscript we did a general description of the number of the exons in expressed circRNAs by comparing the sets of conserved circRNAs and species-specific circRNAs (Figure 2F). To further investigate this, we compared the distribution of number of exons between conserved and species-specific circRNAs revealing that, in general, conserved circRNAs are composed of fewer exons than species-specific circRNAs (p-value = 2.23 x 10^-20^; Wilcoxon rank-sum) (Figure 2—figure supplement 4C).

Yet, as the reviewer points it out we did not explicitly test if conserved circRNAs have a higher proportion of circRNA species with 2 to 3 exons than species-specific circRNAs. This analysis found that conserved circRNAs have a higher proportion of circRNAs with 2 to 3 exons in comparison to species-specific circRNAs (p-value = 4.173173 x 10^-08^ Fisher exact test).

To describe this analysis we have added the following text to the results:

“Anaylsis of the exonic structure of conserved circRNAs, showed that conserved circRNAs contain fewer exons (Figure 2F; Figure 2—figure supplement 4C, p = 2.23 x 10^-20^ Wilcoxon rank sum test) with a significant enrichment to contain 2-3 exons (p-value = 4.17 x 10^-08^ Fisher exact test), which is in line with observations from previous studies (Ragan et al., 2019).”

Therefore, we can conclude that conserved circRNAs have fewer exons with a greater likelihood to be composed of 2 to 3 exons than species-specific circRNAs.

a.8. On page 18 the authors defined relative TPM values. However, the definitions of circRNAsRead and GeneRead are not clear.

We apologize for not stating this clearly. We have added the following section to the methods:

“circRNAs expression of conserved, tissue-conserved, and non-conserved circRNAs was calculated using relative TpMs. […] *Gene TpM* refers to the TpM value of the gene with the exons of the circRNA and *Gene Reads* refers to the number of reads of the gene with the exons of the circRNA.”

b) Further contextualization of the findings. Relevant aspects of the authors' findings have been previously reported and should be properly cited and discussed, includingb.1.- the reports by Rybak et al. (Mol Cell 2015), Veno et al. (Genome Biol. 2015) and You et al. (Nature Neuro 2015, which is not even cited) described that brain circRNAs are conserved, as well as several of the features described for these circRNAs in the manuscript.

We would argue results build on these results, as we compare groups of circRNAs (conserved vs species-specific) whereas these studies compare conserved circRNAs to background distributions. However, we apologize for not more clearly referencing these papers and have now added them to the Results section:

“Initial analysis of conserved circRNAs revealed enrichment within neural tissues with over 70% showing consistent tissue expression across 30 million years of evolution (Supplementary File 2), in line with previous observations (Rybak-Wolf et al., 2015; Venø et al., 2015; You et al., 2015).”

b.2. consider the orientation of the inserted transposons (as showed in Chen LL et al., Cell 2014).

In our original analysis, we considered the orientation of the inserted transposons. We have amended the results to more clearly reflect this (See also our response a.4):

“This includes multiple features previously associated with conserved circRNAs, such as inverted-repeat Alu elements (Jeck et al., 2013; Zhang et al., 2014).”

and

“Introns adjacent to conserved circRNAs also exhibited a significant enrichment for repeat elements (Figure 3D, all p < 1x10^-5^, BH-Wilcox vs species-specific) in particular inverted-repeat L1 and AluJ retrotransposons (Figure 3D, L1: p < 1.22x10^-23^| AluJ: p < 1.48x10^-18^, BH-Wilcox).”

b.3. Extension of downstream introns and inefficient cleavage and polyadenylation are well described as factors modulating exon circularization (see Liang et al. Mol Cell 2017, Ashwall et al., Mol Cell 2014).

We have added these references to the paper in the Results section:

“This includes multiple features previously associated with conserved circRNAs, such as inverted-repeat Alu elements (Jeck et al., 2013; Zhang et al., 2014), as well as exon and intron length (Ashwal-Fluss et al., 2014; Ivanov et al., 2015; Jeck et al., 2013; Liang et al., 2017)”

and in the discussion:

“Both groups are related in the cis- and trans-regulatory features that correlate with circRNA formation such as evidence of recent transposons insertion and extended adjacent introns (Ashwal-Fluss et al., 2014; Ivanov et al., 2015; Jeck et al., 2013; Liang et al., 2017). […] This decreased diversity of conserved circRNA production and increased relative expression is in line with data from linear splicing (Baek and Green, 2005; Barbosa-Morais et al., 2012; Gueroussov et al., 2017; Irimia, Rukov, Roy, Vinther, and Garcia-Fernandez, 2009; Merkin et al., 2012) and suggests circRNA selection is occurring.”

b.4. In the abstract the authors state that "many primate genes produce non-coding circular RNAs". This statement is not precise as it is not clear how many of the circRNAs are coding and how many non-coding. The non-coding should be eliminated. Moreover, seems like the authors don't even know about the research showing that some circRNAS are translated.

We have removed references to “non-coding” in the abstract and elsewhere in the manuscript.

b.5. In line 41, the authors stated that "Back-splicing occurs co-transcriptionally" but it has been shown that back-splicing might occur not only co-transcriptionally (Ashwall Fluss et al. Mol Cell 2014) but both co-transcriptionally and post-transcriptionally (Zhang et al., Cell Rep 2016, PMID: 27068474).

We agree with the reviewers that (back-)splicing can occur both co-transcriptionally, as well as post-transcriptionally. We have corrected this in the text.

b.6. The most important model of this paper is that insertion of young transposon into downstream introns results in longer intronic regions, which delay the RNA polymerase II to the next splice site and increase the possibility of circRNA conservation. However, it has been reported that "Pol II accelerates dramatically while transcribing through genes, but slows at exons" (Jonkers et al., eLife 2014) and "back-splicing outcomes correlate with fast RNA Polymerase II elongation rate" (Zhang et al., Mol Cell 2016). In this case, longer introns might lead to fast, but not slow RNA polymerase II for circRNAs.

We thank the reviewer for drawing our attention to these papers. We agree with the reviewer that RNA polymerase II slows down at exons. However, in this study, we are considering two orthologous intronic regions that differ in length due to the insertion of transposons.

In Zhang et al., they demonstrated that genes containing circRNAs have a faster elongation rate of RNA polymerase II. Our results do not contradict this result, as we are considering the two orthologous intronic regions with orthologous introns and not different genes.

Our model is based on two factors (i) that RNA polymerase slows down when transcribing repeat regions (relative to non-repeat regions) (ii) the time available for back-splicing will be greater in a longer intron than a shorter intron.

For the first factor, there are previous reports showing that the speed of RNA polymerase II decreases when transcribing repetitive elements (e.g. Veloso et al. 2014, Genome Research). The second point is that if you increase the size of the intron by 2 or 4 fold by the insertion of repeat elements it will increase the time prior to the transcription of the next splice site, compared to a shorter intron. This is supported by the relative speeds of RNA Polymerase elongation rate. For example, in Zhang et al., the median rates of elongation were 2.90 and 2.29kb/min for circRNA genes and non-circRNA genes, respectively. Given we identify a two-fold lengthening of the intron in human vs lemur, even an increased rate of RNA polymerase II wouldn’t have a major impact on the relative speed in which the RNA polymerase reaches the next canonical exon in the two orthologous introns. We acknowledge we haven’t described this clearly in the discussion and have therefore added the following section:

“The extension of the final intron therefore increases the likelihood of circRNA formation in time and space. […] This model conforms with previous observations of the enrichment of inverted-repeat Alu elements and of long introns surrounding circRNAs (Ashwal-Fluss et al., 2014; Dong, Ma, Chen, and Yang, 2017; Ivanov et al., 2015; Jeck et al., 2013; Liang and Wilusz, 2014; Rybak-Wolf et al., 2015; Zhang et al., 2014).”

b.7. This manuscript mainly showed the insertion of LINE for circRNA expression and conservation. However, previous studies have suggested the importance of SINE elements, especially Alu elements of primates, in circRNA biogenesis (for example, Jeck et al., RNA 2013, PMID: 23249747; Zhang et al., Cell 2014, PMID: 25242744; Dong et al., RNA Biol 2017, PMID: 27982734). Did the authors observe that Alu is less involved in primate circRNA expression than LINE? Or, since Alu is prevalent among primates, their contribution to conserved circRNAs is less important than LINE?

We apologize for not stating our results more clearly. In our analysis, we replicate previous findings of the importance of Alu elements in our comparisons of all circRNAs vs background. LINE elements display importance when comparing conserved circRNAs vs species-specific circRNAs. Our results, therefore, are in line with the previous literature. However, we agree this was not clearly described and have therefore added the following sentence with the citations requested to the Results section:

“This includes multiple features previously associated with conserved circRNAs, such as inverted-repeat Alu elements (Jeck et al., 2013; Zhang et al., 2014), as well as exon and intron length (Ashwal-Fluss et al., 2014; Ivanov et al., 2015; Jeck et al., 2013; Liang et al., 2017).”

and in the discussion:

“This model conforms with previous observations of enrichment of inverted-repeat Alu elements and observation of long introns surrounding circRNAs (Ashwal-Fluss et al., 2014; Dong, Ma, Chen, and Yang, 2017; Ivanov et al., 2015; Jeck et al., 2013; Liang and Wilusz, 2014; Rybak-Wolf et al., 2015; Zhang et al., 2014).”

and

“Similarly, we identify the adjacent introns of circRNAs are significantly longer with inverted Alu repeats (Ashwal-Fluss et al., 2014; Ivanov et al., 2015; Jeck et al., 2013; Liang et al., 2017), however only in the conserved group do we observe a bias towards lengthening of the downstream adjacent intron with inverted L1 repeats dominating.”

b.8. In line 39, the authors cited Guo et al., Cell 2020 to show as examples of functional circular RNAs especially in the immune and nervous systems, but it has nothing to do with circular RNAs.

We sincerely apologize for this mistake. We referenced the wrong Ling Ling Chen Cell paper. We meant to reference Liu et al. 2019. Many thanks for spotting this error.

b.9. In line 44, Quaking was not likely to be suggested to facilitate "these (assumedly inverted repeat elements) RNA-RNA interactions", instead, ADAR (Ivanov et al., Cell Rep 2015, PMID: 25558066; Rybak-Wolf et al., Mol Cell 2015, PMID: 25921068), DHX9 (Aktaş et al., Nature 2017, PMID: 28355180), NF90/110 (Li et al., Mol Cell 2017, PMID: 28625552) were suggested to be involved in inverted repeat element RNA-RNA interactions to regulate circular RNA biogenesis.

Quaking has been identified as a regulator of RNA-RNA interactions that promotes circRNA interactions (e.g. Conn et al. 2015 Cell). We do agree that NF90/110 and ADAR are important potential regulators and should be included. We have therefore added this gene alongside Quaking.

“…such as Quaking, ADAR, or NF90/110 to facilitate circRNA formation(Conn et al., 2015; Ivanov et al., 2015; Li et al., 2017).”

b.10. In line 67, the authors cited Pipes et al., Nucleic Acids Res 2013 to suggest their used datasets in this study. However, many samples originally in Pipes et al., Nucleic Acids Res 2013 had only mRNA-seq, without total RNA-seq Total RNA-seq datasets for many those samples were updated in Peng et al., Nucleic Acids Res 2015, which should be referenced.

Many thanks for the correction. We will cite both studies in order to support the work of this consortium

c) Revising causality claims based upon associative correlations:c.1. In line 101 the authors state "Many orthologous genes consistently express circRNAs even of the precise back-spliced junction is not conserved implicating importance of trans-factors in controlling cirRNA formation". There is no logic in this statement, long introns and random insertion of repetitive elements in inverse orientation could explain this w/o invoking any trans-acting factor.

We agree. We have corrected this sentence accordingly. It now reads:

“Many orthologous genes consistently express circRNAs even if the precise back-spliced junction is not conserved”.

c.2. The authors claim that "Anaylsis of the exonic structure of conserved circRNAs, showed that conserved circRNAs contain fewer exons, and rarely overlap with other circRNAs (Figure 2G, p = 4:08. 10*64, Fisher exact test; see Methods) displaying back-splicing at unique 5´- and 3´-splice sites. This indicates that these conserved circRNAs possess unique cis- or trans-regulatory features that enable a tight control of the number of exons within a circRNA and the back-spliced junctions used." This is not necessarily like that and the 2 aspects could be concurrent/codependent on a different factor (i.e. intron length).

We agree. We have corrected this sentence accordingly.

“This indicates a tight control of the number of exons within a circRNA and the back-spliced junctions used.”

c.3. After looking for predictive genomic features for circRNA biosynthesis the authors conclude that: "This indicates a core set of 24 cis- and trans-regulatory features drive the conserved formation of circRNAs compared to our background set of introns" This is factually and conceptually wrong, as opredictive value does not mean causality. So likely most of these factors are co-occurring, which is still interesting but should not be overstated.

We agree. We have corrected this sentence accordingly.

“This identifies a core set of 24 cis- and trans-regulatory features enriched within the conserved formation of circRNAs compared to our background set of introns (Figure 3A and 3B). This includes multiple features previously associated with conserved circRNAs, such as inverted-repeat Alu elements (Jeck et al., 2013; Zhang et al., 2014), as well as exon and intron length (Ashwal-Fluss et al., 2014; Ivanov et al., 2015; Jeck et al., 2013; Liang et al., 2017).”